# Structural basis of mitochondrial membrane bending by the I–II–III$_2$–IV$_2$ supercomplex

Alexander Mühleip[1,6,10], Rasmus Kock Flygaard[1,7,10], Rozbeh Baradaran[1,8], Outi Haapanen[2], Thomas Gruhl[3], Victor Tobiasson[1,8,9], Amandine Maréchal[3,4], Vivek Sharma[2,5] & Alexey Amunts[1✉]

Mitochondrial energy conversion requires an intricate architecture of the inner mitochondrial membrane[1]. Here we show that a supercomplex containing all four respiratory chain components contributes to membrane curvature induction in ciliates. We report cryo-electron microscopy and cryo-tomography structures of the supercomplex that comprises 150 different proteins and 311 bound lipids, forming a stable 5.8-MDa assembly. Owing to subunit acquisition and extension, complex I associates with a complex IV dimer, generating a wedge-shaped gap that serves as a binding site for complex II. Together with a tilted complex III dimer association, it results in a curved membrane region. Using molecular dynamics simulations, we demonstrate that the divergent supercomplex actively contributes to the membrane curvature induction and tubulation of cristae. Our findings highlight how the evolution of protein subunits of respiratory complexes has led to the I–II–III$_2$–IV$_2$ supercomplex that contributes to the shaping of the bioenergetic membrane, thereby enabling its functional specialization.

Mitochondrial energy conversion requires an electron transport chain (ETC) that generates a proton motive force across the inner mitochondrial membrane to drive the essential adenosine triphosphate (ATP) formation by F$_1$F$_o$-ATP synthase. The ETC consists of four multisubunit membrane complexes: complex I (CI; NADH:ubiquinone (UQ) oxidoreductase), complex II (CII; succinate:UQ oxidoreductase), complex III (CIII; cytochrome $bc_1$) and complex IV (CIV; cytochrome $c$ oxidase). Biochemical and structural analyses have shown that these components can organize into supercomplexes containing CI, CIII dimer (CIII$_2$) and CIV[1,2]. CII transfers electrons from succinate via its covalently bound flavin adenine dinucleotide (FAD) and iron–sulfur clusters to UQ and is also a component of the tricarboxylic acid cycle, making a functional link between the two central metabolic pathways[3]. Although CII has been suggested to interact with mammalian ETC complexes[4–8], it was not experimentally found as part of any characterized supercomplex. In addition, for the bioenergetic process to occur, a specific topology of the crista membranes that form functionally distinct high-potential compartments is critical[9]. An established mechanism for maintenance of such a topology relies on oligomerization of ATP synthase and its specific interplay with lipids[10–14]. In ciliates, the inner mitochondrial membrane is organized as tubular cristae, which was previously explained by the helical row assembly of ATP synthase[12,15,16].

We purified the intact respiratory supercomplex from mitochondria of the ciliate protist *Tetrahymena thermophila* and determined its structure by single-particle cryo-electron microscopy (cryo-EM) (Extended Data Fig. 1 and Supplementary Table 1). At an overall resolution of 2.9 Å,

the structure revealed that CI, CII, CIII$_2$ and a CIV dimer (CIV$_2$) associated into a 5.8-MDa supercomplex (Fig. 1). When viewed along the membrane plane, the assembly of more than 300 transmembrane helices displays a bent shape, indicating that the accommodating membrane adopts a local curvature with a radius of approximately 20 nm (Fig. 1). Masked refinements resolved individual structures that together form an assembly of 150 different protein subunits and 311 bound lipids (Extended Data Figs. 1 and 2a–f and Supplementary Tables 1 and 2). CIV$_2$ is associated with the long side of the membrane region of CI, opposite CIII$_2$. This arrangement is markedly different compared with known mammalian supercomplexes[17,18] and correlates with the acquisition of four ciliate-specific CI subunits that would clash with the position of CIV as seen in mammals (Fig. 1d and Extended Data Fig. 3). CII is anchored between CI and CIV, highlighting the unique architecture and composition of the native supercomplex (Fig. 1). In-gel activity assays confirmed co-migration of functional CI, CII and CIV in a high-molecular-mass band, which we assigned to the intact supercomplex (Extended Data Fig. 2g; and see source data for Extended Data Fig. 2).

CIV$_2$ is the most divergent of the four ETC complexes (Extended Data Figs. 4 and 5). We modelled 105 lipids, four UQs and 53 protein chains per monomer, of which four are mitochondrially encoded. We found that two of those subunits, previously annotated as ciliate-specific Ymf67 and Ymf68 (also known as COX3) represent complementary protein fragments, with coding genes split in the mitochondrial genome by a tRNA$^{Trp}$ gene insertion (Extended Data Fig. 6). Each fragment is extended by over 400 and 200 residues, respectively. Together, they

[1]Science for Life Laboratory, Department of Biochemistry and Biophysics, Stockholm University, Solna, Sweden. [2]Department of Physics, University of Helsinki, Helsinki, Finland. [3]Institute of Structural and Molecular Biology, Birkbeck College, London, UK. [4]Institute of Structural and Molecular Biology, University College London, London, UK. [5]HiLIFE Institute of Biotechnology, University of Helsinki, Helsinki, Finland. [6]Present address: School of Infection and Immunity, University of Glasgow, Wellcome Centre for Integrative Parasitology, Glasgow, UK. [7]Present address: Department of Molecular Biology and Genetics, Danish Research Institute of Translational Neuroscience-DANDRITE, Nordic EMBL Partnership for Molecular Medicine, Aarhus University, Aarhus C, Denmark. [8]Present address: MRC Laboratory of Molecular Biology, Cambridge, UK. [9]Present address: National Center for Biotechnology Information, National Library of Medicine, National Institute of Health, Bethesda, MD, USA. [10]These authors contributed equally: Alexander Mühleip, Rasmus Kock Flygaard. ✉e-mail: amunts@scilifelab.se

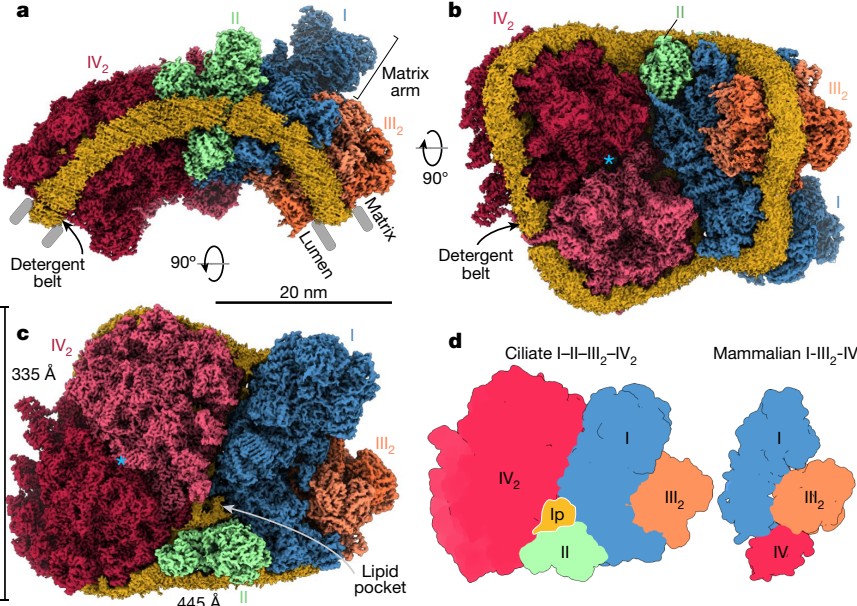

**Fig. 1 | The supercomplex contains all four ETC components. a**, Side view of the supercomplex density showing the curved detergent micelle (yellow). **b**, Lumenal view illustrates how complexes CI, CII and CIV$_2$ stabilize each other. The blue asterisk indicates the symmetry axis of CIV$_2$. **c**, Matrix view shows CII binding in a wedge between CI and CIV$_2$, resulting in the enclosure of a lipid pocket (lp). The blue asterisk indicates the same position as in **b**. **d**, Architecture comparison of the ciliate I–II–III$_2$–IV$_2$ supercomplex (this study; left) with the mammalian I–III$_2$–IV respirasome (Protein Data Bank (PDB) accession 5J4Z; right), highlighting a different location of CIV$_2$ that is correlated with acquisition of CI subunits that stabilize CIII$_2$.

form a functional COX3 (COX3a and COX3b), including the conserved seven transmembrane helix fold (Extended Data Fig. 6e). In our structure, COX3a and COX3b extend throughout the CIV membrane region, and COX3b has evolved interactions with CI subunits on the matrix side, thereby mediating the supercomplex assembly (Fig. 2a,b). In particular, COX3b forms a contact with a peripheral amphipathic helix of NDUCA1, which is part of a zinc-free γ-carbonic anhydrase (γ-CA) heterotrimer (Fig. 2c). The γ-CA heterotrimer was previously reported in viridiplantae and ciliates (both diaphoretickes)[8,19] and our structure demonstrates that it acts as a structural scaffold within the supercomplex architecture. Another CI–CIV contact at the same site involves COXTT2 with a N-terminal globin-like domain that interacts with NDUFA3 and was not resolved in the individual CIV$_2$ structure[8], suggesting that it becomes ordered to mediate supercomplex formation (Fig. 2a,c). The second major interaction site at the CI–CIV interface, which is on the lumenal side of the membrane, also involves a fragmented protein subunit, this time from CI. Consistent with the observation with respect to the protein splitting in CIV, we modelled the N-terminal extension of the ND5 fragment (ND5a), as well as the newly identified protein subunit NDUTT16 (from CI) (Fig. 2a,b,d). NDUTT16 engages in interactions with at least four subunits of CIV, as well as an interfacial CIV haem group (Fig. 2d and Extended Data Fig. 5f).

Our finding of the split core subunits gaining a capacity of establishing intercomplex contacts to stabilize the supercomplex that curves the membrane suggests a putative mechanism by which gene fragmentation, followed by its expansion, might convey subunit function, which was also observed in mitoribosomes[20,21]. Overall, the CI–CIV$_2$ interface involves 25 subunits, forming an extensive buried interface of approximately 2,300 Å$^2$ with a curved membrane region (Fig. 2a,b), and thus a single mutation is unlikely to disrupt the curvature induction.

*T. thermophila* CII binds in a wedge-shaped gap formed by CI–CIV in our structure (Fig. 1a–c). In addition to the four canonical subunits (SDHA–D), it is composed of 11 ciliate-specific subunits SDHTT1–11 (Fig. 2e and Extended Data Fig. 7). The matrix module SDHA and SDHB forms a conserved head region, containing both the covalently bound FAD and the three iron–sulfur clusters (Extended Data Fig. 7b).

The membrane anchor is formed by two small subunits, SDHC (7 kDa) and mitochondria-encoded SDHD (5 kDa), which could only be assigned by locating topologically conserved transmembrane helices in the map (Extended Data Fig. 7a,c,d). At the lumen, an approximately 70-kDa module (SDHTT2, SDHTT4, SDHTT5 and SDHTT8) anchors CII to CI–CIV (Fig. 2e,f and Extended Data Fig. 7a). SDHTT5 interacts with a helix of NDUTT5 protruding from the membrane arm of CI, and with the Surf1-like protein subunit COXTT8 (CIV) (Fig. 2a,e). At the same position, COXTT8 interacts with CI via the N-terminal ND5a extension, and with Ymf75, COXTT27 and COXTT18 at the CIV dimer interface. Thus, the three complexes—CI, CII and CIV—are connected in the lumen (Fig. 2f).

Between SDHB and SDHD, we identified a ligand, which we assign to ubiquinone bound to the conserved proximal Q$_p$ site[22,23] (Extended Data Fig. 7e). On the matrix side, the 36-kDa soluble subunit SDHTT1 contains a bis-histidine C-type haem covalently bound by a single cysteine residue (Extended Data Fig. 7a). Although it is exposed to the membrane region, at a distance of approximately 60 Å from the Fe$_3$S$_4$ cluster, the non-canonical haem C is located too far to participate in direct CII electron transfer (Extended Data Fig. 7a). To provide further evidence of the presence of additional haem groups in the supercomplex, we recorded visible absorption redox spectra of the purified sample (Extended Data Fig. 8. Deconvolution of the merged absorption bands characteristic of B-type and C-type haems clearly revealed the contribution of at least one additional haem group in the supercomplex, in addition to those present in CIII, with specific absorption at 556 nm.

The presence of a functional ETC with CII is consistent with previous observations that *T. thermophila* can utilize succinate to drive cellular respiration[24]. Our native structure with bound CII, which contributes to the ubiquinol pool, demonstrates that assembly of the supercomplex is not limited to the proton-pumping respiratory complexes (CI, CIII and CIV). Beyond decreasing cytochrome *c* transfer distance[25], this suggests a potential role of supercomplex formation in mediating increased UQ diffusion, as suggested in analogous membrane systems with high protein to lipid ratios[26,27]. Furthermore, the tubular membrane morphology may require anchoring of CII to the curved supercomplex

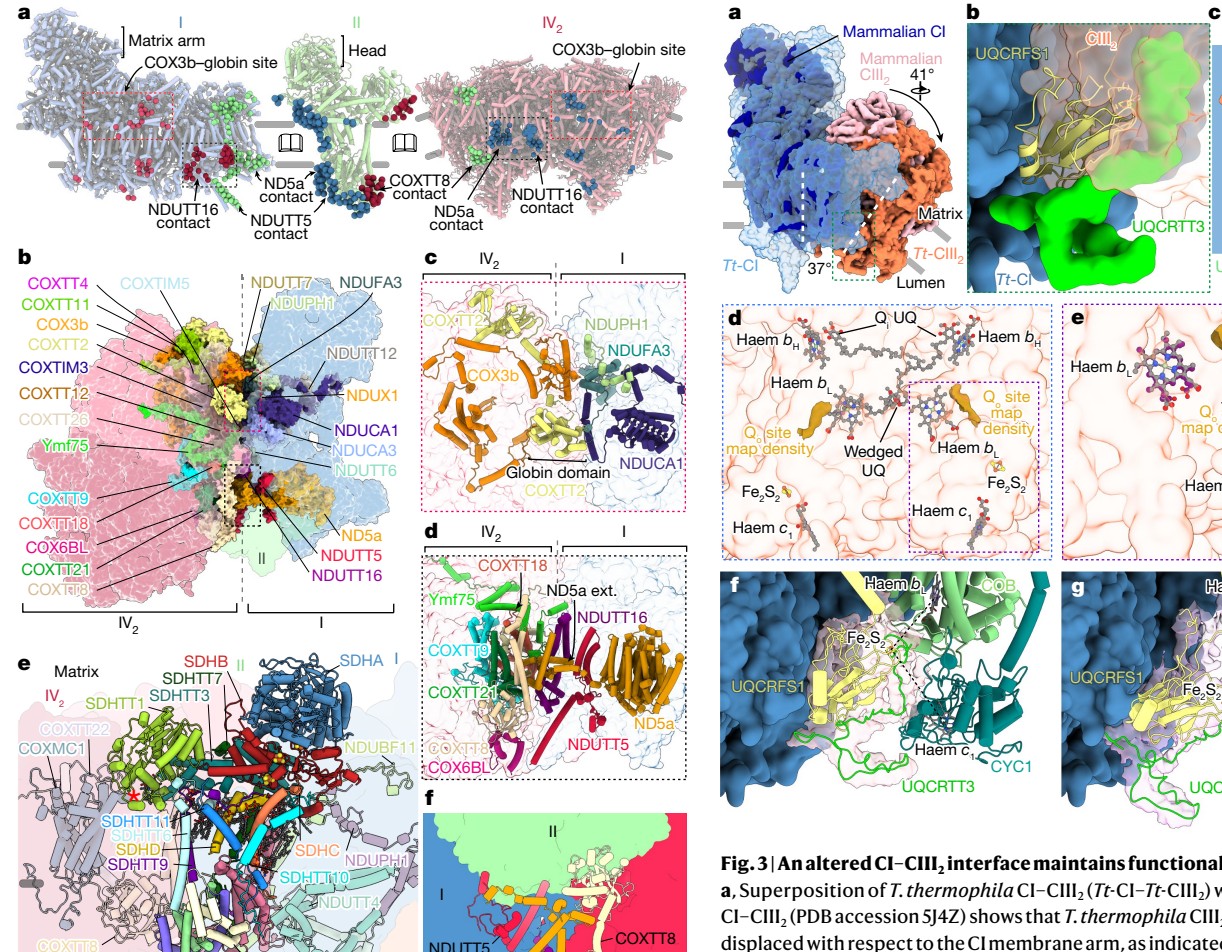

**Fig. 2 | The CI–CIV association and binding of CII. a**, Open-book view with contact sites of CI with CII–CIV$_2$ (left), CII with CI–CIV$_2$ (middle) and CIV$_2$ with CI–CII (right). Interactions are shown as spheres (CI in blue, CII in green, CIV$_2$ in red or dark pink). Only one CIV monomer interacts with CII. The main interaction sites are indicated. **b**, The CI–CIV$_2$ interacting subunits are shown in coloured surfaces. **c**, The COX3b–NDUCA1 contact site. **d**, The NDUTT16–ND5a contact site. ext., extension. **e**, CII binding to subunits of CI–CIV (transparent). The red asterisk marks C-type haem. **f**, CI, CII and CIV are connected via the membrane and lumenal regions.

**Fig. 3 | An altered CI–CIII$_2$ interface maintains functional symmetry of CIII$_2$. a**, Superposition of *T. thermophila* CI–CIII$_2$ (*Tt*-CI–*Tt*-CIII$_2$) with mammalian CI–CIII$_2$ (PDB accession 5J4Z) shows that *T. thermophila* CIII$_2$ is tilted, rotated and displaced with respect to the CI membrane arm, as indicated by the white dashed lines and arrow, due to acquisition of new proteins. The green box view is shown in **b**. **b**, Tilted *T. thermophila* CIII$_2$ results in an interaction between UQCRFS1 and the CI membrane arm together with UQCRTT3. **c**, *T. thermophila* CIII$_2$ shows a membrane-accessible tunnel extending to the COB Q$_i$ sites. The blue box view is shown in **d**. **d**, The Q$_i$ UQs are located close to haem $b_H$, with one UQ wedged in between the two haem $b_L$ molecules. The density map of *T. thermophila* CIII$_2$ shows two features corresponding to Q$_o$ sites. The purple box view is shown in **e**. **e**, The Q$_o$ site density overlaps with ubiquinol in the bovine CIII$_2$ structure (PDB accession 1NTZ). **f**,**g**, 3D maps showing B state (**f**) and C state (**g**) conformations of UQCRFS1 and UQCRTT3. Only COB and CYC1 proteins are shown for clarity, highlighting distances of Fe$_2$S$_2$ to haem $c_1$ and $b_L$ (black dashes).

to retain it in the functionally relevant cristae and prevent diffusion to flat membrane regions.

CIII$_2$ in our structure is tilted with respect to CI by 37° (Fig. 3a). This tilted arrangement offsets the transmembrane region, consistent with its curved membrane environment. The interface involves 20 subunits and 19 bound native lipids interacting within the matrix, transmembrane and lumenal domains (Extended Data Fig. 9). When compared with the mammalian counterpart, CIII$_2$ is rotated by 41° and shifted approximately 14 Å due to acquisition of four CI subunits, as well as the CIII subunit UQCRTT1 (refs. [17,28–30]) (Fig. 3a and Extended Data Fig. 9a–c). This arrangement results in a specific CI–CIII$_2$ contact with one copy of the Rieske iron–sulfur protein of CIII (UQCRFS1) interacting with the CI membrane arm (Fig. 3b). The interaction site is further augmented by a hitherto unidentified protein UQCRTT3, which interacts with the lumenal head domain of UQCRFS1 and wedges in between the interface to CYC1 (Fig. 3b).

We traced the membrane-accessible UQ tunnel, lined by UQCR10 and ciliate-specific subunit UQCRTT2 (Fig. 3c), leading to the conserved COB haem $b_H$, where the density for a bound (semi)-UQ was

observed in the Q$_i$ site[31] (Fig. 3d and Extended Data Fig. 9e,f). Furthermore, we observed map density features close to the two conserved haem $b_L$ groups, which probably correspond to ubiquinols bound at the Q$_o$ sites (Fig. 3d,e and Extended Data Fig. 9e,f). The distances between the Q$_o$ site, haem $b_L$ and haem $b_H$ within each CIII monomer are consistent with those observed in mammalian CIII$_2$, with the two haem $b_L$ in COB being bridged by a non-canonical UQ. We detected density for two copies of the flexible UQCRFS1 head domain (Extended Data Fig. 9g), which contrasts with recent work that found only the head domain proximal to the CI quinone tunnel to display flexibility, whereas the distal domain at the CI interface was proposed to be non-functional in electron transport[8]. Using focused 3D classification for the distal UQCRFS1 head domain, we then identified two classes probably representing the extremes of the head domain movement: from the B state where the Fe$_2$S$_2$ cluster is distanced from haem $c_1$ to the C state where the Fe$_2$S$_2$ cluster is closest[32,33] (Fig. 3f,g and Extended Data Fig. 9i). This movement of the UQCRFS1 head domain is coupled to conformational changes in the unidentified UQCRTT3 protein, suggesting a potential role for this subunit in regulation of CIII$_2$ activity.

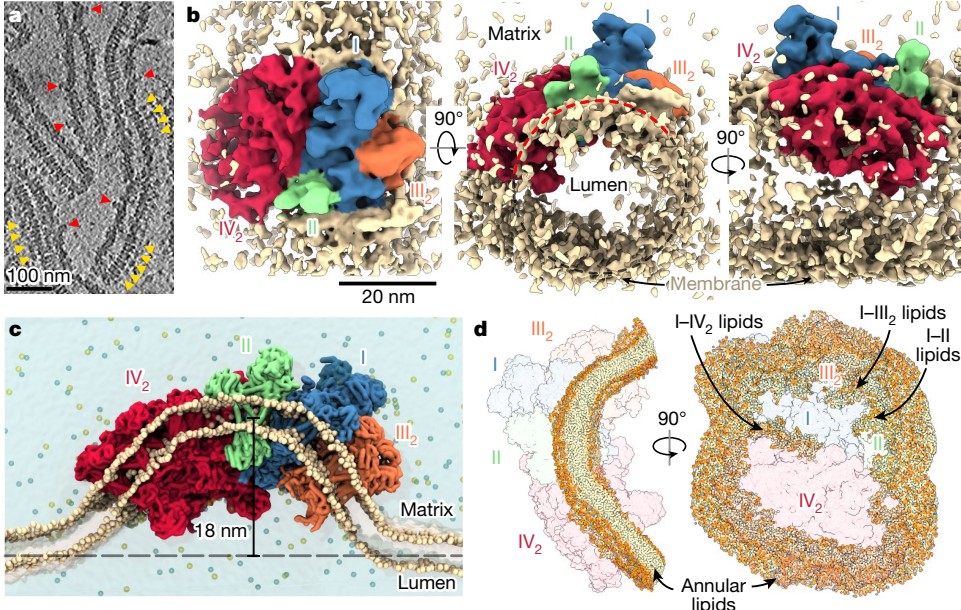

**Fig. 4 | In situ structure and of the I–II–III₂–IV₂ supercomplex indicate a membrane-bending function. a**, Cryo-tomographic slice of tubular cristae with a diameter of approximately 40 nm. ATP synthase and the supercomplex are marked with yellow and red arrowheads, respectively. **b**, Subtomogram average of the I–II–III₂–IV₂ supercomplex revealing a preferred orientation in tubular membranes and an arc-shaped structure subtending approximately 130° (red dashes). **c**, Coarse-grained molecular dynamics simulation showing that the arched membrane region of the supercomplex generates a significant membrane curvature, resulting in an 18-nm local displacement of the membrane from the bilayer plane. **d**, Molecular dynamics simulation reveals the curved structure of the annular lipid shell surrounding the supercomplex, as well as lipid-filled subcomplex interfaces.

Thus, our observation of the distal UQCRFS1 head domain flexibility, together with the haem $b_L$-wedged UQ, suggests that functional symmetry is maintained in the ciliate CIII₂ despite deviation from the structural symmetry.

To investigate whether the membrane-bending capacity of the supercomplex is biologically relevant, we performed cryo-electron tomography of isolated mitochondrial membranes. Cryo-tomograms revealed approximately 40-nm tubular cristae densely packed with helical ATP synthase rows and supercomplexes, identified by the conspicuous CI matrix arm (Fig. 4a). To elucidate the supercomplex architecture in situ, we performed subtomogram averaging and obtained a map at 28 Å resolution (Extended Data Fig. 1e). The subtomogram average confirmed the presence of the supercomplex, which fits our atomic model (Extended Data Fig. 1f), suggesting that this is the most abundant form. The appearance of a tubular membrane density in the subtomogram average suggests that the supercomplex adopts a preferred orientation, with its CI–CIV₂ interface approximately aligned with the long axis of the tube (Fig. 4b). Furthermore, the curved membrane region of the supercomplex subtends an angle of approximately 130°, indicating that it contributes to the tubular shape of the cristae.

To elucidate the membrane-shaping activity of the supercomplex, we performed coarse-grained molecular dynamics simulations. When placed into a planar lipid bilayer, the supercomplex induces a curved membrane topology, displacing the membrane by 18 nm from the original plane (Fig. 4c and Supplementary Video 1), in contrast to the protein-free lipid bilayer, which remains relatively flat (Supplementary Video 2). Furthermore, the annular lipid shell surrounding the complex in the equilibrated system displays a highly curved architecture, supportive of an active role in membrane curvature induction (Fig. 4d and Supplementary Video 3). In addition, we observed lipid pockets in the transmembrane interfaces between subcomplexes, which suggests that their maintenance is crucial for the integrity of the supercomplex (Extended Data Fig. 2a,b), as was also shown for photosynthetic membranes[34]. Molecular dynamics simulations of CIV₂ in a membrane suggest that it can also induce membrane bending in its immediate vicinity,

similar to the supercomplex (Extended Data Fig. 10a,b). However, for the formation and stability of a tubular architecture, juxtapositioning of individual subcomplexes is probably necessary. This is also supported by the increased arc length of the detergent belt observed in the cryo-EM structure of the supercomplex, when compared with CIV₂ (Extended Data Fig. 10b). Finally, molecular dynamics simulations of the supercomplex lacking CII showed a similar wrapping of the lipid bilayer as for the full supercomplex, with lipids filling the generated void (Extended Data Fig. 10c). This indicates that CII association may contribute to complex stability while retaining the enzyme in the tubular cristae regions, where ubiquinol is required as a substrate.

Our results indicate that cristae shaping involves both the respiratory supercomplex and the ATP synthase that together generate membrane tubulation. Although the coiled ATP synthase rows fix the helix diameter at 130 nm (refs. [12,15]), supercomplexes serve the function of confining a narrow crista diameter of around 40 nm, which allows tight packing of cristae, thereby increasing the surface area of the bioenergetic membrane. This membrane-shaping organization of the respiratory supercomplex is markedly different from the mammalian homologue, which resides in the flat crista regions. Furthermore, because every crista represents an independent functional compartment[35], restriction of the crista diameter by the respiratory supercomplex probably serves to minimize the volume, thereby potentially contributing to higher local concentration of electron carriers, ensuring that proton translocation results in an increased local proton motive force, ultimately optimizing conditions for ATP synthesis. This is consistent with the observation that mutant yeast strains with large, balloon-like cristae display respiratory defects[36,37]. Thus, our findings show how respiratory supercomplexes together with other factors can organize the architecture of the bioenergetic membrane, providing a mechanism for enabling its functional specialization.

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

# Methods

## Purification of *T. thermophila* supercomplex

*T. thermophila* cells were grown at 36 °C and harvested as previously described[12,20], and cell pellets were resuspended in homogenization buffer (20 mM HEPES-KOH pH 7.5, 350 mM D-mannitol, 5 mM EDTA and 1x protease-inhibitor tablet) and lysed in a Dounce homogenizer on ice. Intact mitochondria were isolated first by differential centrifugation of the lysate; following membrane solubilization, the lysate was cleared at 30,000*g* for 30 min at 4 °C, and finally, on a discontinuous sucrose gradient with 15%, 23%, 32% and 60% w/v sucrose in buffer SEM (20 mM HEPES-KOH pH 7.5, 250 mM sucrose and 1 mM EDTA) at 14,1371*g* for 60 min at 4 °C in an SW28 rotor. Intact mitochondria sedimented to the interface between 32% and 60% sucrose and were collected from the gradient, snap-frozen in liquid nitrogen and stored at −80 °C. The isolated mitochondria were lysed in buffer A (25 mM HEPES-KOH pH 7.5, 25 mM KCl, 5 mM MgCl$_2$ and 4% w/v digitonin) for 1 h on ice. This procedure was previously confirmed as a gentle solubilization method. Following mitochondrial membrane solubilization, cleared lysate was placed on a sucrose cushion (25 mM HEPES-KOH pH 7.5, 25 mM KCl, 5 mM MgCl$_2$, 0.1% w/v digitonin and 30% w/v sucrose) in Ti70 tubes and centrifuged at 164,685*g* for 1 h. The pellet was gently washed and finally resuspended in buffer D (25 mM HEPES-KOH pH 7.5, 25 mM KCl, 5 mM MgCl$_2$ and 0.1% w/v digitonin). Before loading sample material on a size-exclusion chromatography column, larger aggregates were pelleted at 30,000*g* for 20 min at 4 °C. Cleared sample was loaded on a Superose 6 Increase 3.2/300 column equilibrated in buffer D, collecting elution fractions of 100 µl throughout the run. Peak fractions were used immediately for cryo-grid preparation.

## UV–visible difference spectroscopy

The sample obtained from the sucrose cushion step was analysed for haem content and supercomplex composition using UV–visible difference spectroscopy and clear native-PAGE (CN-PAGE) combined with in-gel activity assays. UV–visible difference spectra were recorded between 390 nm and 675 nm using a home-built spectrophotometer. Protein samples were diluted as necessary in 50 mM HEPES and 0.1% digitonin at pH 8.0. Spectra were measured from sodium dithionite-reduced minus air-oxidized spectra. When multiple absorption bands overlapped, spectra were deconvoluted using the peak analysis function in OriginPro 2015 (OriginLab Corporation).

## Gel electrophoresis

NativePAGE 3 to 12%, Bis-Tris, 1.0 mm, Mini Protein Gel (Invitrogen) pre-cast gels were used for CN-PAGE. The gels were loaded with a protein ladder (NativeMark, Invitrogen) and four identical sample lanes where the protein samples were mixed with NativePAGE Sample Buffer (final concentration, 50 mM BisTris, 6N HCl, 50 mM NaCl, 10% w/v glycerol and 0.001% Ponceau S, pH 7.2) as per the manufacturer's instruction. Electrophoresis was conducted at 4 °C, first at 150 V for 30 min with NativePage Light Blue Cathode buffer (50 mM BisTris, 50 mM Tricine, pH 6.8, and 0.002% Coomassie G-250) and then at 250 V for 150 min with NativePAGE Anode Buffer (50 mM BisTris and 50 mM Tricine, pH 6.8). In-gel activity assays were performed following published protocols[38]. In brief, each sample lane from CN-PAGE was incubated with an aqueous solution to reveal (1) protein bands (0.02% Coomassie G-250, overnight) or the presence of active (2) CI (2 mM Tris pH 7.4, 2.5 mg ml$^{-1}$ nitrotetrazolium blue chloride (NBT) and 0.1 mg ml$^{-1}$ NADH, 15 min), (3) CII (5 mM Tris pH 7.4, 2.5 mg ml$^{-1}$ NBT, 84 mM succinic acid and 0.2 mM phenazine methosulfate, 30–40 min) and (4) CIV (0.05 mM KPi pH 7.4, 0.5 mg ml$^{-1}$ 3,3′-diaminobenzidine (DAB) and 1 mg ml$^{-1}$ cytochrome *c* from *Saccharomyces cerevisiae*, overnight). The reactions were stopped by incubation in 10% (v/v) acetic acid, followed by multiple exchanges of water. The gels shown in this paper are representative of two experiments from two separate supercomplex preparations.

## Cryo-EM sample preparation and data collection

Supercomplex eluted at a concentration of approximately 10 mg ml$^{-1}$. Aliquots of the peak fraction were diluted in buffer D to 0.75 mg ml$^{-1}$ before applied to cryo-grids. Quantifoil R2/2-300 grids floated with a homemade 3-nm amorphous carbon layer were glow-discharged immediately before applying a 3 µl sample. Grids were vitrified using liquid ethane cooled by liquid nitrogen in a Vitrobot Mark IV, with a 30-s wait time before blotting grids for 3 s at blot force of 0. A total of 26,063 movies were collected on a Titan Krios (Thermo Fisher Scientific) operated at 300 kV at a nominal magnification of ×165,000 (0.83 Å per pixel) with a Quantum K2 camera (Gatan) using a slit width of 20 eV. An objective lens aperture of 70 µm was used, with an exposure rate of 4.26 electrons per pixel per second with 5-s exposure fractionated into 20 frames, and the defocus range was 0.6–2.6 µm.

## Cryo-EM data processing

Motion correction was performed in the internal implementation of RELION-3.1 (ref. [39]), followed by contrast transfer function estimation by CTFFIND4. Initial rounds of particle picking and 2D classification, followed by ab initio reconstruction, 3D classification and preliminary refinement of the supercomplex. Template-based particle picking in RELION was then used to pick and extract 1,664,103 particles. 2D classification and 3D heterogeneous refinement steps in cryoSPARC v2 (ref. [40]) were then used to separate supercomplex particles from copurified ATP synthase, resulting in a final 138,746 supercomplex particles used for subsequent refinement. Following a consensus refinement in cryoSPARC, per-particle contrast transfer function refinement and Bayesian polishing were performed in RELION-3.1. For final refinements in cryoSPARC, particles were downsampled from a 724-pixel box to 480 pixels, resulting in a pixel size of 1.25 Å per pixel. Masked refinements of the respective supercomplex subregions resulted in map resolutions of 2.9 Å for the entire supercomplex, 2.8 Å for CI, 3.0 Å for CII, 2.8 Å for CIII and 2.6 Å for CIV$_2$. Reported map resolutions are according to gold-standard Fourier shell correlation using the 0.143-criterion. To assess flexibility of the Rieske subunit wedged in the CI–CIII$_2$ interface, we performed focused 3D classification in RELION-3.1 using pre-aligned particles with a mask on the extended area around the headgroup of the Rieske subunit. Classification into ten classes resulted in maps confirming flexibility of the structural element, with two classes corresponding closely to the previously reported B state and C state (Fig. 3f,g).

## Cryo-electron tomography and subtomogram averaging

Crude mitochondrial pellets were resuspended in an equal volume of buffer containing 20 mM HEPES-KOH pH 7.4, 2 mM EDTA, 250 mM sucrose and mixed in a 1:1 ratio with 5-nm colloidal gold solution (Sigma Aldrich) and vitrified as described above on glow-discharged Quantifoil R2/2 Au 200 mesh grids. Tilt series were acquired on a Titan Krios operated at 300 kV with a K3 camera (slit width of 20 eV) using SerialEM or the EPU software (Thermo Fisher Scientific). Mitochondrial membranes were imaged at a nominal magnification of ×42,000 (2.11 Å per pixel) and an exposure rate of 19.5 electrons per pixel per second with a 3 e$^-$ Å$^{-2}$ exposure per tilt fractionated into five frames, with tilt series acquired using the exposure-symmetric scheme[41] to ±60° tilt and a 3° tilt increment. Following motion correction in motionCor2, tomographic reconstruction from the tilt series was performed in IMOD[42] using phase-flipping and a binning factor of 2. Tomograms were contrast enhanced using nonlinear anisotropic diffusion filtering to facilitate manual particle picking of supercomplex particles based on the matrix arm of CI. Subtomogram averaging was performed in PEET[43]. Initial references were generated from the data by averaging after rotating subvolumes into a common orientation with respect to the membrane based on manually assigned vectors. Following initial rounds of averaging to generate a suitable reference, data were manually split into half-sets and refined independently, following low-pass

filtering to 50 Å. Averaging of 360 particles from 12 tomograms resulted in a 28 Å subtomogram average.

## Model building and refinement

Manual model building was performed in Coot[44] and new subunits identified directly for the cryo-EM map. For identified canonical subunits, homology models were generated using SWISS-MODEL. Bound cardiolipins were unambiguously identified from their head group density. Other natively bound lipids were tentatively modelled as phosphatidylcholine, phosphatidylethanolamine or phosphatidic acid based on head group densities. Real-space refinement of atomic models was performed in PHENIX using secondary structure restraints[45]. Atomic model statistics were calculated using MolProbity[46].

Given the mild solubilization conditions we used, for CIII$_2$, the cryo-EM map showed density located on the pseudo-$C_2$ symmetry axis between the two COB haem $b_L$ molecules, displaying planar map features consistent with the quinone moiety of UQ. The density clearly indicates that UQ can bind in two orientations, related by the symmetry rotation of the dimer. In either orientation, the quinone moiety is positioned close to a haem $b_L$, where it potentially could accept electrons for transfer across the dimer axis. In the recent amphipol CIII$_2$ structure[8], the isoprenoid tail of UQ was modelled in the equivalent position; however, the planar density for the quinone was missing. This orientation-equivalent binding of UQ between the two COB haem $b_L$ molecules, together with the B-state and C-state Rieske conformations, suggest a maintained functional symmetry of ciliate CIII$_2$ within the supercomplex.

In CI, we identified 49 canonical subunits and 21 subunits that we assign as phylum-specific. In each CIV monomer, we identified 11 subunits homologous to mammalian CIV (COX1, COX2, COX3a, COX3b, COX5B, COX6A, COX6B, COX6C, COX7A, COX7C and NDUFA4) and 42 ciliate-specific subunits, most of which are peripherally associated around the mitochondrial protein core. Three of the mammalian subunits missing in *T. thermophila* CIV (COX4, COX7B and COX8) are at the interface where two mitochondrial carriers are bound. The mitochondrially encoded core subunit COX3 is split into two fragments. Most of the TM helices are contributed by the C-terminal of COX3b, which is encoded by the mitochondrial *ymf68* gene. The newly annotated Ymf68 is structurally conserved, apart from the missing helix (H1), which is structurally replaced by Ymf67. We therefore assign *ymf67* and *ymf68* of the ciliate mitochondrial genome as separately encoding the COX3a and COX3b subunit fragments, respectively. On the *T. thermophila* mitochondrial genome, *ymf67* and *ymf68* genes are located on the same strand, but are separated by the gene encoding tRNA$^{Trp}$, suggesting that a transposition event may have led to the fragmentation of the original gene encoding COX3. tRNA genes are known to be among the most motile elements in metazoan mitochondrial DNA. Both COX3a and COX3b fragments have evolved substantial subunit extensions that thread through the augmented CIV monomer unit to recruit lineage-specific subunits and mediate supercomplex assembly.

In the CIV dimer, the dimer interface of 17,000 Å$^2$ is dominated by 16 species-specific subunits. Furthermore, when aligned on the CIV core, a comparison of the mammalian and ciliate structures reveals that the two dimers display markedly different architectures, dimer axes and distances between COX1 cores. This suggests that the dimerization of the ciliate CIV probably evolved through the acquisition of lineage-specific subunits and reflects the constraints of the unique tubular membrane environment.

In addition, each CIV monomer complex contains two different Surf1-like proteins, which were reported to complement defects causing Leigh syndrome in humans. In our structure, two Surf1-like proteins are permanently attached to CIV and display similar overall structures, consisting of a lumen-exposed soluble domain and a TM helix hairpin. The two Surf1 proteins are facing each other, bound on opposite sides of each CIV core.

The presence of subunit extension and accessory subunits in CIV generates a pronounced cavity around the cytochrome $c$-binding site. However, overlaying of a *T. thermophila* cytochrome $c$ homology model suggests that the canonical binding site is not obstructed. Cytochrome $c$ binding is known to be driven by electrostatic interactions with the CuA domain of COX2, which forms a negatively charged patch in mammals. This structural feature is positively charged in *T. thermophila*, interacting mainly with H1 of cytochrome $c$, which displays a flipped polarity. We conclude that the experimentally observed functional incompatibility of *T. thermophila* CIV and mammalian cytochrome $c$ is not due to divergent architecture, but to an inverted surface charge of the binding pocket.

## Molecular dynamics simulations

We performed coarse-grained (CG) molecular dynamics simulations on the entire *T. thermophila* supercomplex structure using Martini3 forcefield[47] to study the rearrangement of the lipid bilayer around the highly bent protein assembly. Using the martinize2 (version 2.6) tool, we transformed the atomistic structure into a CG model (atoms clustered into Martini beads)[48]. Using the small molecules database and the existing topologies of phospholipids available in the Martini3 forcefield[47], we generated the force-field parameters for cardiolipin. Cofactors and resolved lipids in the structure were not included in the simulated model system, and only protein was simulated to study the dynamics of lipid molecules around it. First, the CG model of the protein structure was minimized for 100 steps in vacuum to remove possible steric clashes. Then, the minimized CG supercomplex was embedded in a large (75 nm × 75 nm) hybrid membrane slab (POPE:POPC:CL in 4:2:1 ratio) using the insane.py script[49]. The CG protein–membrane system was solvated using standard Martini3 water beads and 100 mM Na$^+$ and Cl$^-$ ions. Starting from this initial position, the simulation system was minimized keeping all beads free, first in double precision to resolve steric clashes between the lipids (maximum of 500 steps) and then in regular single precision (maximum of 10,000 steps). After minimization, with 4,000 kJ mol$^{-1}$ nm$^{-2}$ harmonic constraints on the backbone beads, the system was equilibrated using velocity-rescaling thermostat[50] and Berendsen barostat[51] for 10 ns. During production runs, the 4,000 kJ mol$^{-1}$ nm$^{-2}$ harmonic constraints on the backbone beads were maintained. Velocity-rescaling thermostat[50] and Parrinello–Rahman barostat[52] were used for temperature (310 K) and pressure (1 bar) control in the production phase. Coulombic interactions were treated with the reaction-field algorithm using $\varepsilon_r = 15$ (ref. [53]). The Verlet cut-off scheme was implemented with a Lennard–Jones cut-off of 1.1 nm (ref. [54]). The time step of the CG molecular dynamics simulations was 20 fs. Initial simulation replicas showed incomplete or unstable wrapping of the membrane around the protein, so we translated the lipid bilayer patch in $z$-direction and altered the insertion angle of the supercomplex to find an initial position that allowed the membrane to equilibrate and wrap fully around the protein (Systems T1-T7, simulation lengths of 0.9–2.8 μs, total of 9.6 μs). After finding the correct insertion of the protein into the membrane, we initiated three independent simulation replicas (Systems P1-P3, simulation lengths of 10 μs each, total of 30 μs). The simulations were performed using the Gromacs software (version 2021)[55]. In addition to molecular dynamics simulations of the entire supercomplex in the lipid bilayer, we also performed molecular dynamics simulations of the pure lipid bilayer (see above; three replicas, 10 μs each) and of CIV$_2$, CI–CIII$_2$ and CI–CIII$_2$–CIV$_2$ subcomplexes (1–3 replicas, 1–5 μs each).

## Data visualization and analysis

Images and videos were rendered using ChimeraX[56] and Visual Molecular Dynamics[57]. To analyse the *T. thermophila* cytochrome $c$-binding site, the mammalian cytochrome $c$-bound CIV structure (Protein Data Bank ID: 5IY5) was overlaid to the *T. thermophila* structure. Using AlphaFold2 (ref. [58]), a *T. thermophila* cytochrome $c$ structure was predicted and overlaid to both the mammalian structure and the *T. thermophila* structures. The composite map of the complete respiratory supercomplex

was generated in ChimeraX[56]. This map was only used for visualization, and not for atomic model refinement; instead, a consensus map was used. The buried areas of the CI–CII–CIII$_2$–CIV$_2$ supercomplex and the CIV$_2$ interface were calculated in ChimeraX[56].

## Reporting summary

Further information on research design is available in the Nature Portfolio Reporting Summary linked to this article.

## Data availability

The atomic coordinates were deposited in the Protein Data Bank (PDB) under accession numbers 8BQS (supercomplex), 8B6F (CI), 8B6G (CII), 8B6H (CIV) and 8B6J (CIII). The cryo-EM maps have been deposited in the Electron Microscopy Data Bank (EMDB) under the respective accession numbers: EMD-16184, EMD-15865, EMD-15866, EMD-15867 and EMD-15868. The subtomogram averages have been deposited under EMD-15900. The atomic coordinates that were used in this study are: 1NTZ (cytochrome $bc_1$), 5IY5 (cytochrome $c$) and 5J4Z (ovine supercomplexes). The full versions of all gels are provided in the source file. An Excel file containing the visible absorption spectroscopy data and their analysis has been added. All the data will be publicly available. Source data are provided with this paper.

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

**Acknowledgements** We thank SciLifeLab EM facility (funded by the KAW, EPS and Kempe foundations), the Center for Scientific Computing, Finland for high-performance supercomputing resources, the EMBO Young Investigator Program, the Swedish Foundation for Strategic Research (FFL15:0325), the Ragnar Söderberg Foundation (M44/16), Cancerfonden (2017/1041), the European Research Council (ERC-2018-StG-805230) and the Knut and Alice Wallenberg Foundation (2018.0080). V.S. was supported by the Academy of Finland, the Sigrid Jusélius Foundation, the Jane and Aatos Erkko Foundation, the Magnus Ehrnrooth Foundation and the University of Helsinki. A.Maréchal was supported by the Medical Research Council (CDA MR/M00936X/1, transition support MR/T032154/1). We also thank M. Kozlov, B. Bruininks, W. Kulig, P. C. T. Souza and M. Girych for discussions on molecular dynamics simulations.

**Author contributions** A. Mühleip, R.K.F. and A.A. designed the project. V.T. performed cell culturing and isolation of mitochondria. A. Mühleip and R.K.F. prepared the sample and collected cryo-EM data. A. Mühleip, R.K.F. and R.B. processed cryo-EM data and built the model. A.Mühleip performed cryo-electron tomography and subtomogram averaging. O.H. and V.S. performed molecular dynamics simulations. T.G., A. Maréchal and A.A. performed biochemical and spectroscopic analyses. A. Mühleip, R.K.F. and A.A. wrote the manuscript with contributions from O.H., V.S. and A.Maréchal. All authors contributed to revising the manuscript.

**Funding** Open access funding provided by Stockholm University.

**Competing interests** The authors declare no competing interests.

**Additional information**
**Correspondence and requests for materials** should be addressed to Alexey Amunts.

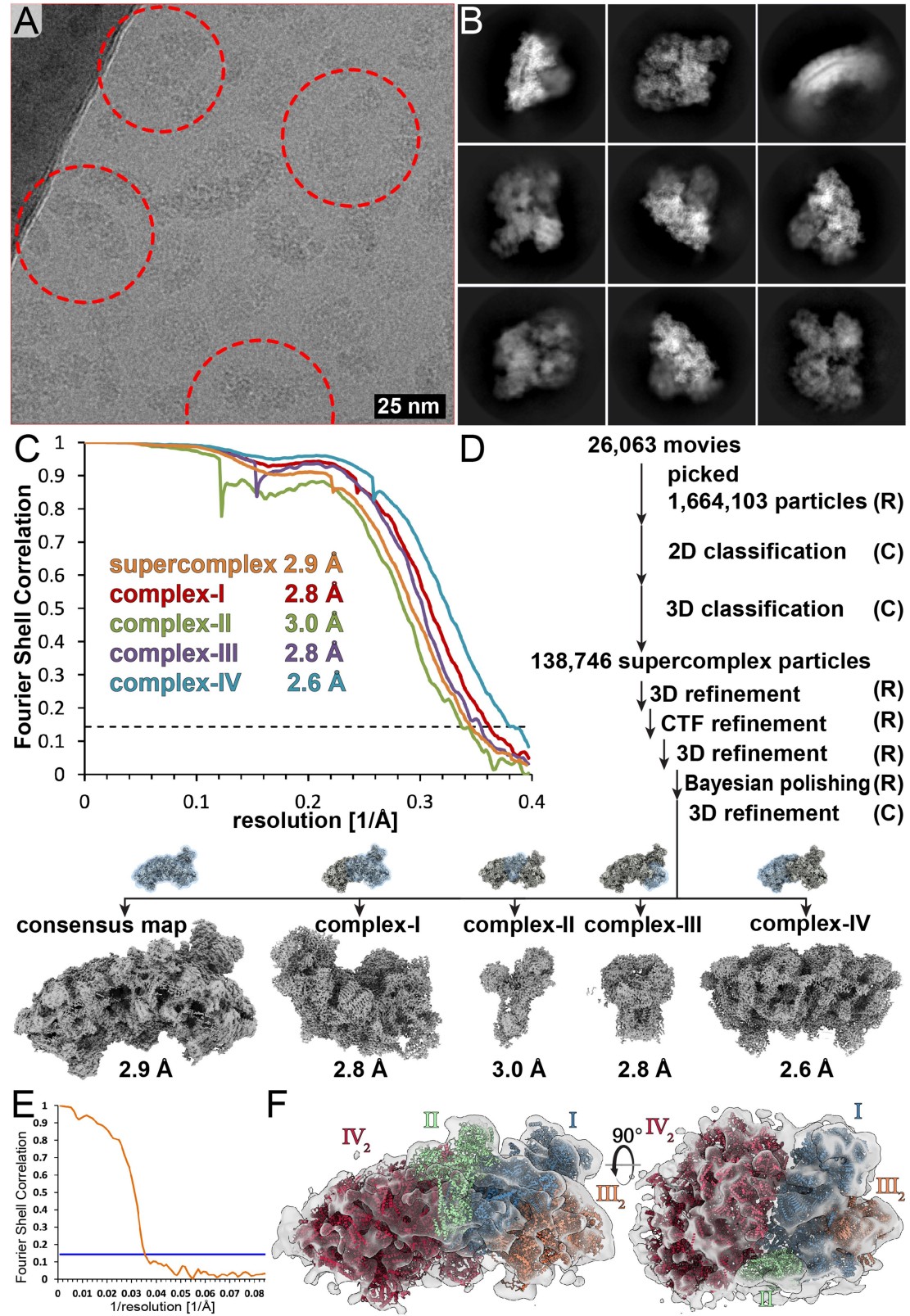

**Extended Data Fig. 1 | Cryo-EM and subtomogram averaging. (A)** A total of 26,063 movies were recorded and analyzed, a representative micrograph is shown. Particles that were not respiratory supercomplex were discarded by classification, since they cannot contribute to reconstruction. Cryo-EM structures were successfully obtained from three preliminary datasets. **(B)** 2D class averages. **(C)** Fourier Shell Correlation according to the 0.143 gold-standard criterion. **(D)** processing workflow in RELION-3.1 (R) and cryoSPARC2 (C). **(E)** Fourier Shell correlation of the subtomogram average indicating a resolution of 28 Å. **(F)** Subtomogram average map-model overlay.

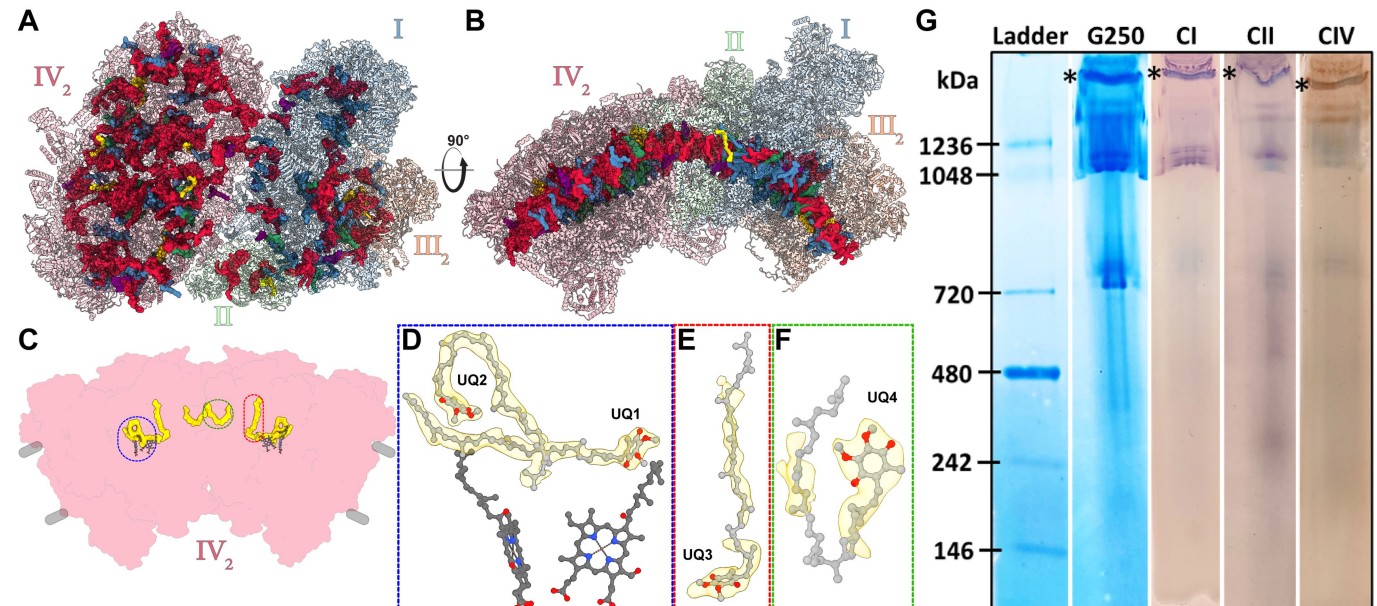

**Extended Data Fig. 2 | Bound native lipids of the ciliate I-II-III$_2$-IV$_2$ supercomplex reveal a curved membrane region.** Top view **(A)** and side view **(B)** with bound lipids. Cardiolipin (red), phosphatidylcholine (blue) phosphatidylethanolamine (green), phosphatidic acid (yellow), ubiquinone-8 (purple). **(C-F)** Location of the four ubiquinone-8 molecules with closeup views **(D-F)**. **(G)** Coomassie-stained CN-PAGE and in-gel activity assay to visualize the purified I-II-III$_2$-IV$_2$ supercomplex. Individual lanes highlight protein bands with active CI, CII (purple) and CIV (brown), respectively. The band marked with an asterisk (*) was tentatively assigned as the intact supercomplex.

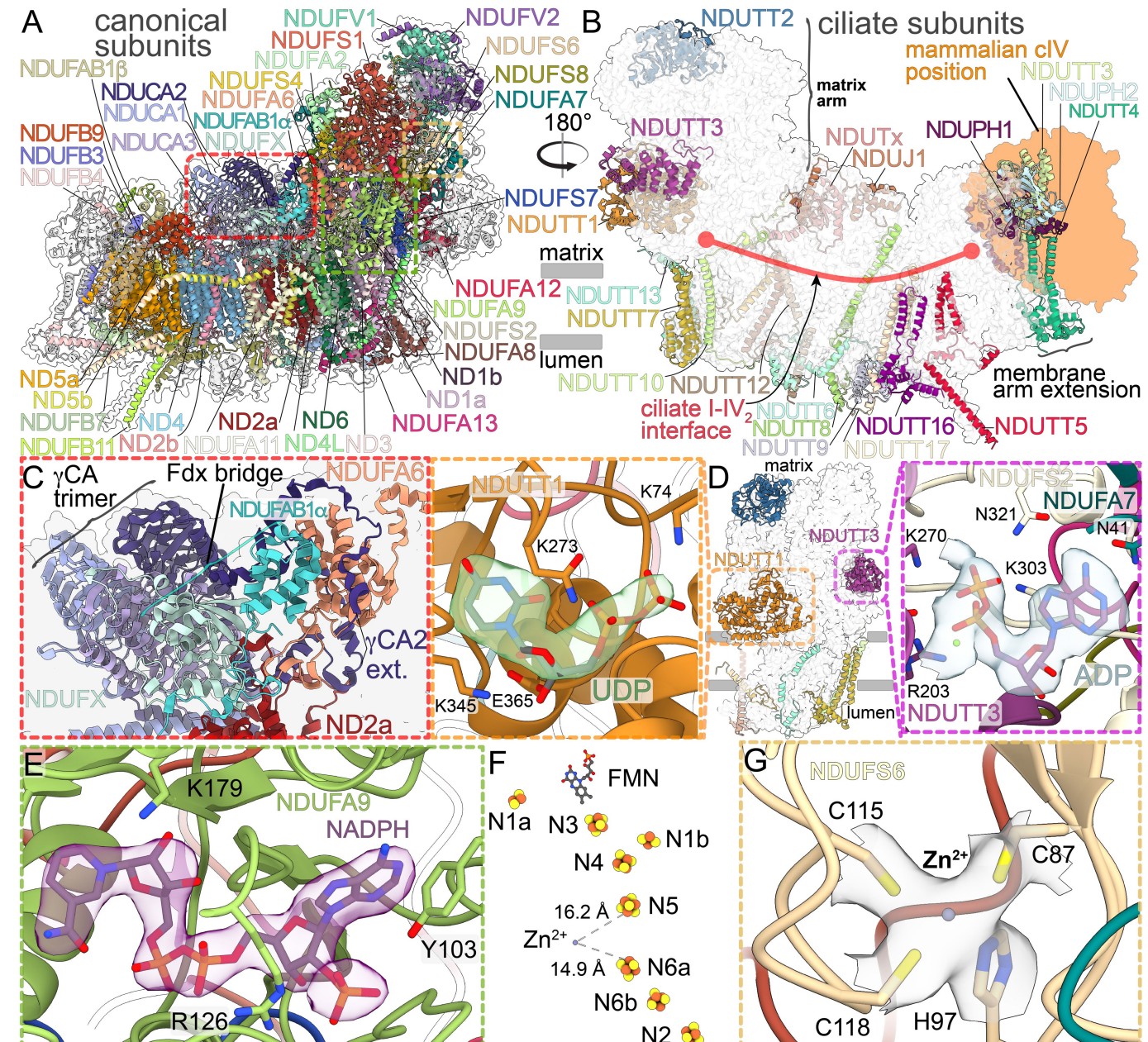

**Extended Data Fig. 3 | Lineage-specific structural features of CI dictate a divergent supercomplex architecture.** (**A**) Canonical subunits of CI (coloured) include conserved antiporter-like folds (ND2a/b, ND4, ND5a/b), and γ-carbonic anhydrase trimer. (**B**) Augmented CI architecture occluding canonical CIV binding site. (**C**) The ferrodoxin bridge of NDUAB1-a and NDUFX connects ND2a of the membrane arm and NDUA6 of the matrix arm. (**D**) The matrix arm contains bound subunits NDUTT1 with UDP and NDUTT3 with ADP. (**E**) NDUFA9 contains a bound NADPH molecule close to the CI Q-tunnel. (**F**) Redox centers in CI matrix arm. (**G**) zinc ion coordinated by NDUS6.

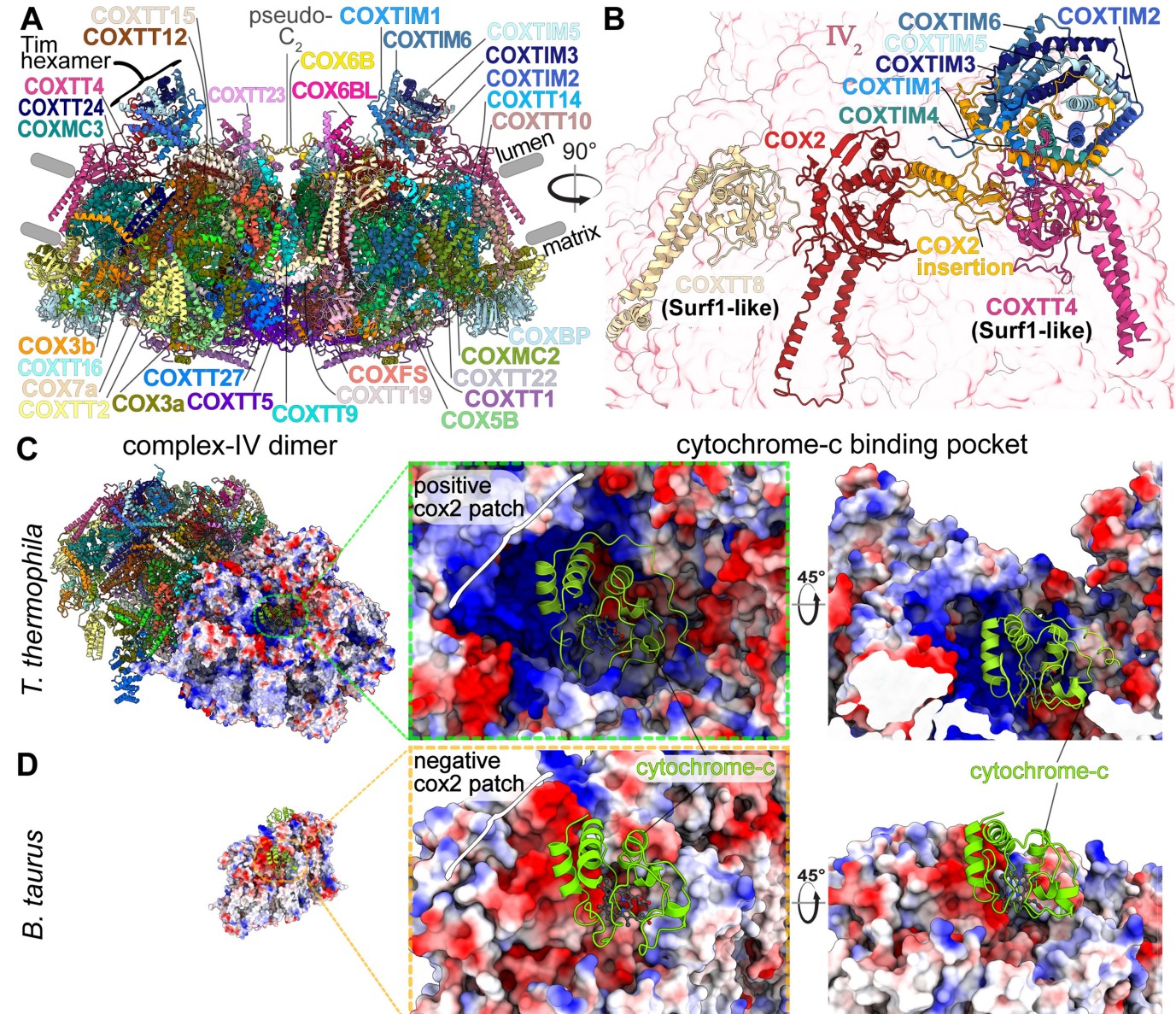

**Extended Data Fig. 4 | CIV₂ contains a bound Tim hexamer and cytochrome-*c* binding site with inverted electrostatic charge. (A)** sideview of CIV₂, containing numerous accessory subunits, including a Tim hexamer. **(B)** Closeup view of COX2, which contains an insertion that recruits the Tim hexamer to the CIV dimer. Furthermore, COX2 interacts with the two Surf1-like proteins COXTT8 and COXTT4. **(C-D)** ciliate and mammalian CIV₂ structures. An overlay of a predicted Tt-cytochrome-*c* structure fits the cytochrome-*c* binding crater without clashes.

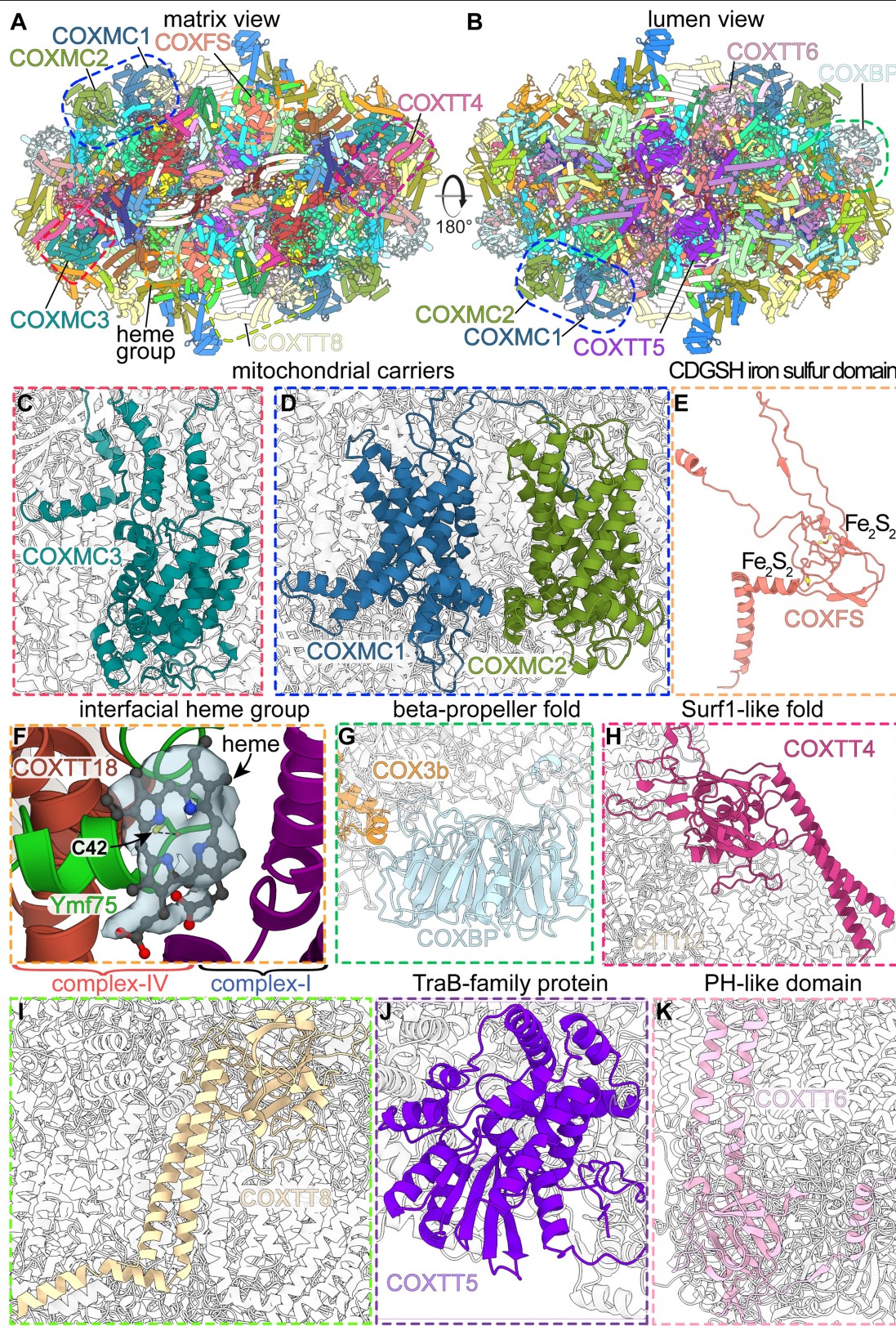

**Extended Data Fig. 5 | The augmented CIV$_2$ contains numerous associated compact-fold subunits. (A,B)** Matrix and lumen view of the CIV dimer, subunit locations of insets are indicated. **(C,D)** Subunits COXMC1, MC2 and MC3 form mitochondrial carrier folds. COXFS is a CDGSH iron sulfur domain **(E)**.

**(F)** Ymf75 coordinates a noncanonical heme group at the CIV periphery. **(G-K)** Compact folds of accessory subunits include a seven-bladed beta-propeller (G), two Surf1-like proteins (H, I), a TraB-family protein (J) and a Pleckstrin homology (PH) like domain (K).

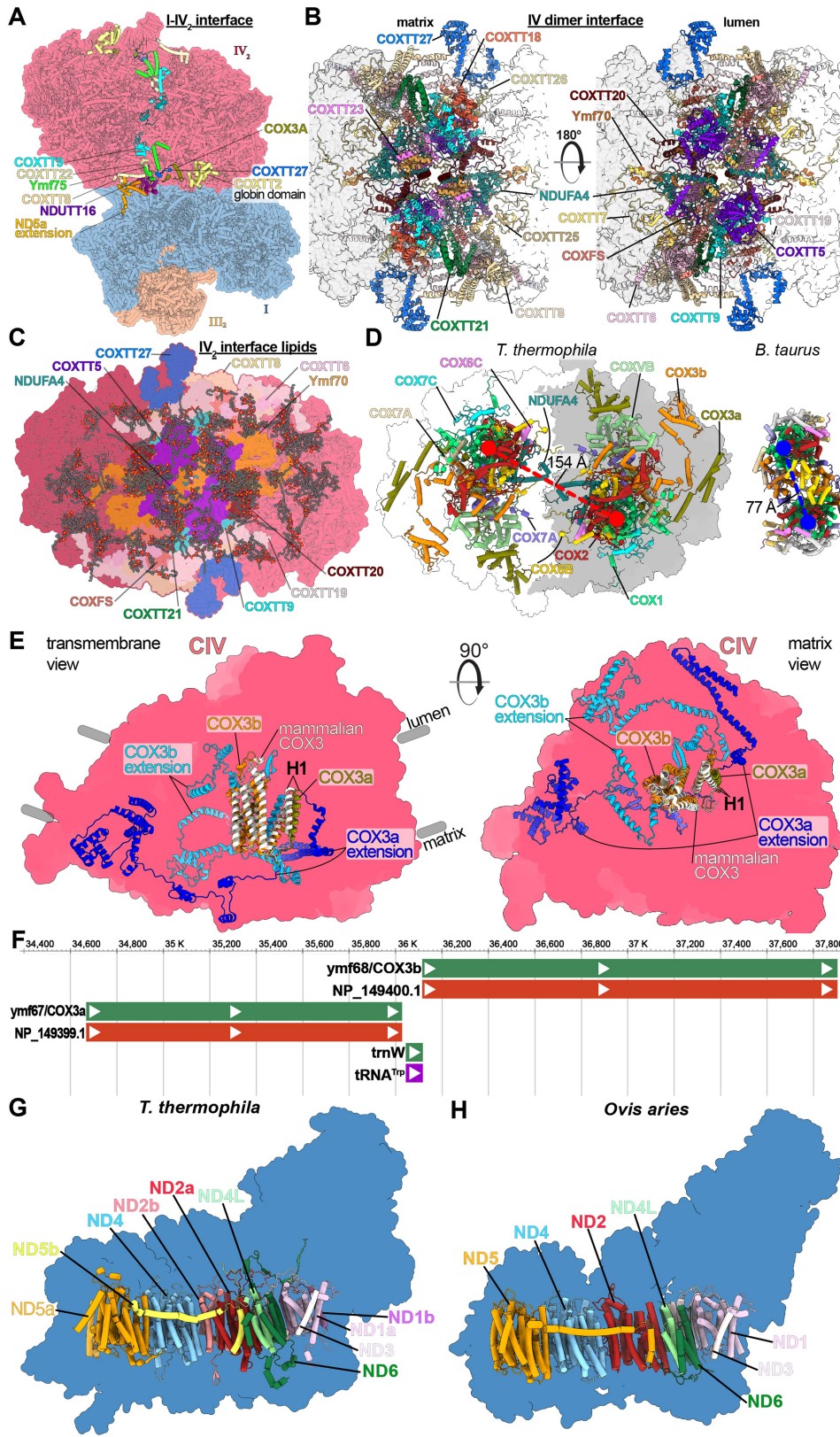

**Extended Data Fig. 6** | See next page for caption.

**Extended Data Fig. 6 | Fragmentation and extension of COX3 and CI proton-pump subunits enables formation of a divergent supercomplex assembly.**
**(A)** The supercomplex displays additional subunits and ordered structural elements (colored cartoons) compared to the amphipol-embedded $CIV_2$ dimer (PDB 7W5Z). In the CI-IV interface this includes subunit NDUTT16 and a globin-like domain of COXTT2. **(B)** Matrix and lumenal views of the $CIV_2$ interface include NDUFA4. **(C)** The $CIV_2$ contains 210 lipids, many of which populate the dimer interface. **(D)** The *T. thermophila* dimer is augmented compared to the bovine dimer and contains the previously unassigned canonical subunit NDUFA4, which is absent in the bovine dimer (right, PDB 3X2Q). Vectors (red, blue) indicate different distances between COX1 centers. **(E)** Overlay of the mammalian COX3 (PDB 5IY5) and the Tt-COX3 (this study) shows that the N-terminal fragment (COX3a), corresponds to Helix-1 (H1) of the conserved structure. COX3b makes up most of the conserved COX3 fold and contains large extensions that contribute to supercomplex formation. **(F)** *T. thermophila* mitochondrial genome region showing the insertion of the tRNA-Trp gene in between *ymf67*/COX3a and *ymf68*/COX3b (genes in green, protein/RNA transcripts in red and purple). **(G)** ciliate CI outline (blue) with the antiporter-like subunits shown as cylinders. The canonical proton pumps ND1, ND2 and ND5 are encoded by split genes, resulting in chains *a* and *b*. **(H)** mammalian CI (blue, PDB 5LNK) with single-chain ND1-6 subunits color-coded as in G.

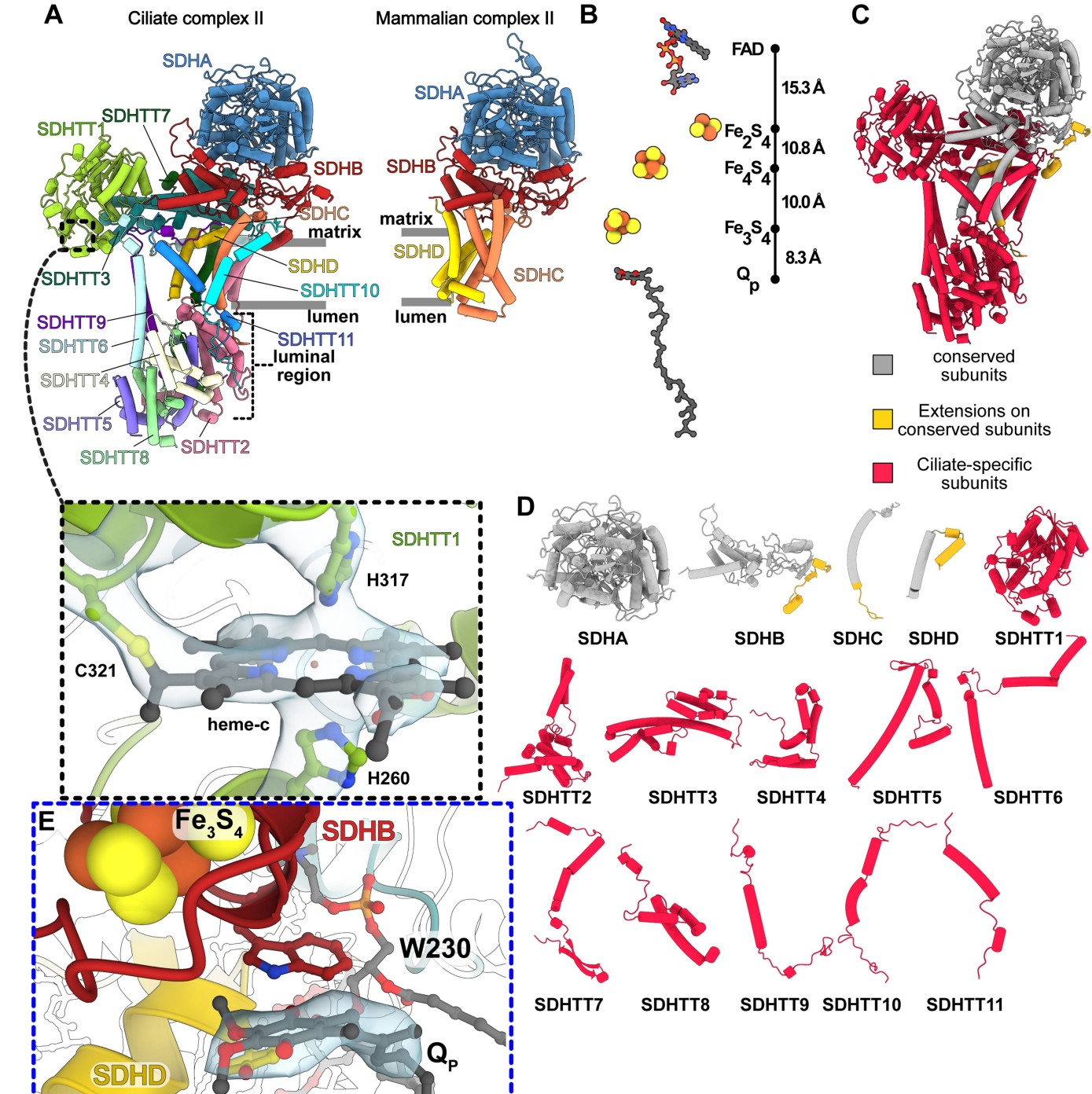

**Extended Data Fig. 7 | Conservation and divergence of ciliate CII. (A)** Ciliate CII (left) colored by subunits and shown in comparison to mammalian CII (right, PDB 1ZOY) with similar color scheme for subunits SDHA-D. SDHC and SDHD are substantially reduced and partially replaced by ciliate-specific subunits. Dashed box highlights a noncanonical cysteine-linked heme C that is coordinated by two axial histidines in SDHTT1. **(B)** Conserved electron transfer chain in CII. **(C)** CII colored by conserved subunits (grey), extensions (gold) and ciliate-specific subunits (red). **(D)** Individual CII subunits with similar color coding as in C. **(E)** CII contains a conserved proximal ubiquinone.

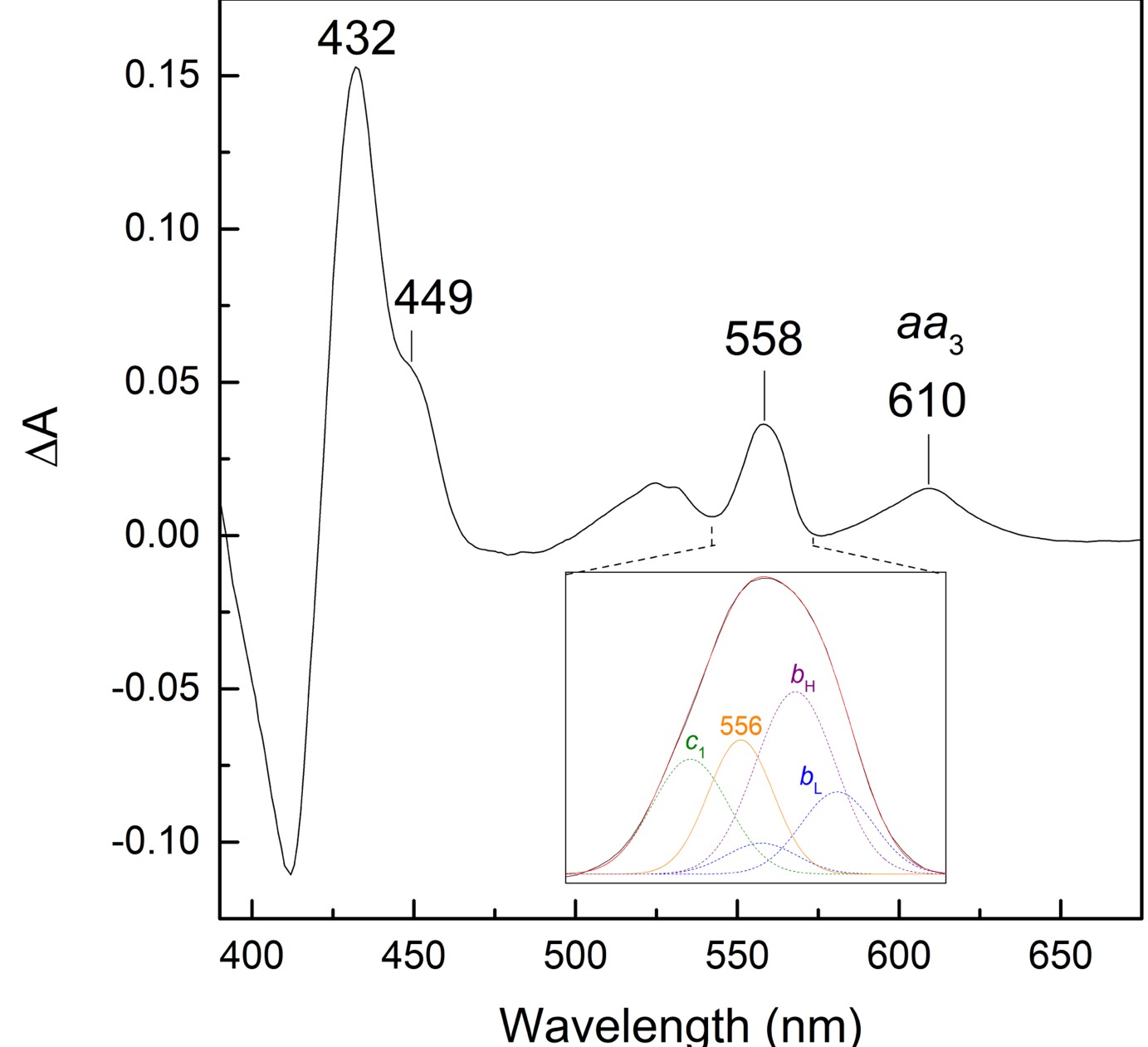

**Extended Data Fig. 8 | UV-visible redox absorption spectrum indicate activity of the supercomplex.** Dithionite minus air-oxidized difference spectrum. Peaks are labeled that correspond to the absorption of the A-type hemes present in CIV (449/610 nm, $aa_3$) as well as merged absorption bands of other B- and C-type hemes (maxima at 432/558 nm). Inset: Deconvolution of the 558-nm absorption band highlighting the contribution of the B- and C-type hemes of CIII ($b_H$ at 561 nm, purple; $b_L$ at 558 and 565 nm, blue; $c_1$ at 552 nm, green) and the presence of at least another heme-protein with absorption maximum at 556 nm (orange). The spectrum shown is representative of two separate supercomplex preparations.

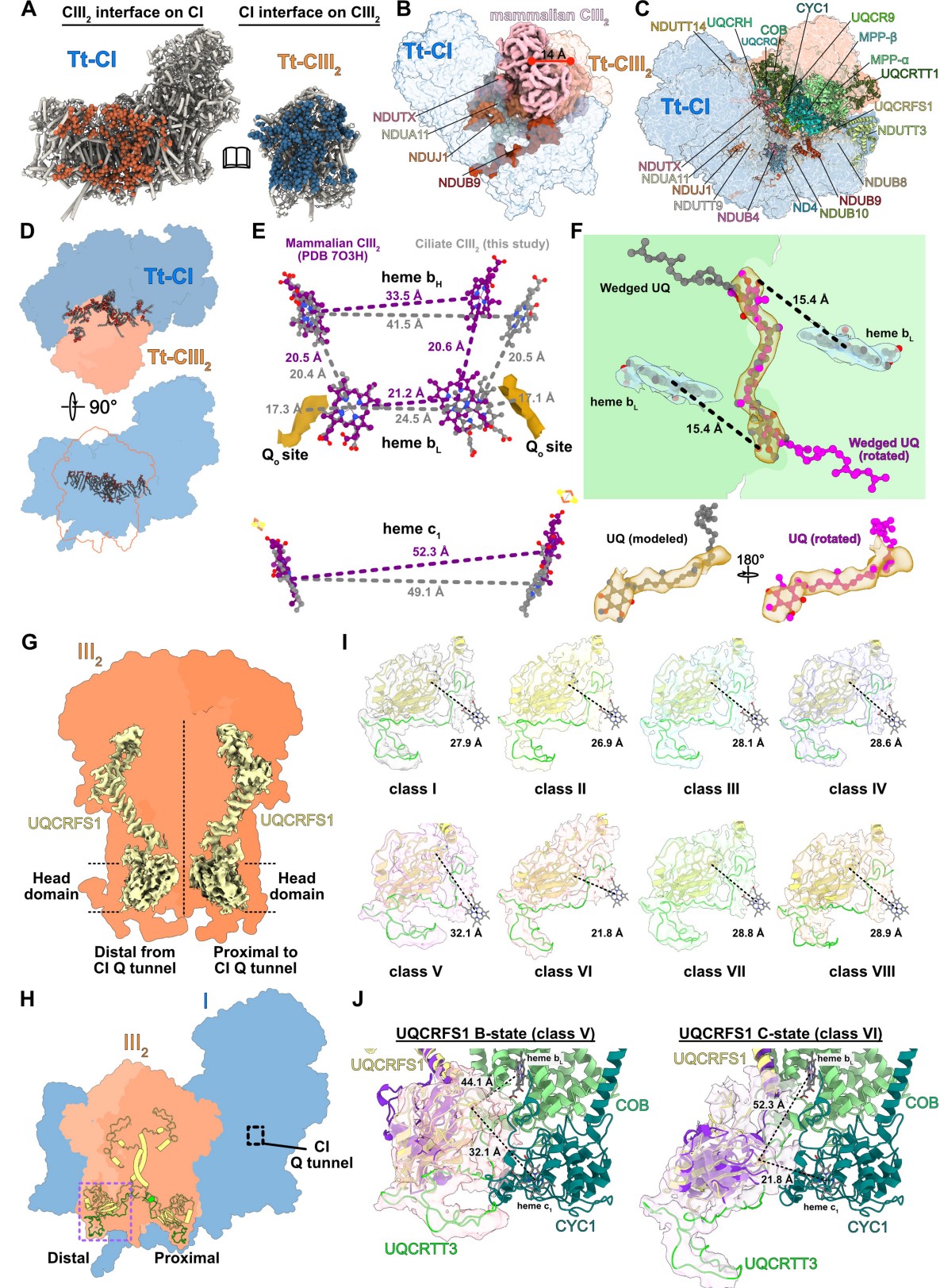

**Extended Data Fig. 9** | See next page for caption.

**Extended Data Fig. 9 | Extensive CI-CIII$_2$ interface involves numerous lipids and accommodates flexible head domain of distal UQCRFS1. (A)** Buried surfaces in the CI-CIII$_2$ interface. CIII$_2$ contacts on CI (orange) and CI contact points on CIII$_2$ (blue; >9,000 Å$^2$ interface). **(B)** Four Tt-CI subunits would clash with mammalian CIII$_2$ in canonical position, thus Tt-CIII$_2$ is displaced 14 Å from Tt-CI. **(C)** Tt-CI (light blue) and Tt-CIII$_2$ (light orange) with interface-forming subunits shown in colored ribbons. **(D)** Orthogonal views of bound native lipids in the I-III$_2$ interface. **(E)** Distances between ciliate (grey) and mammalian (purple) heme $b_H$, heme $b_L$ and heme $c_1$ groups (measuring from Fe atoms) and the ciliate Q$_o$ sites. Superposing mammalian CIII$_2$ (PDB 7O3H) shows that ciliate heme $b_H$ and heme $b_L$ are displaced further away from the symmetry-related hemes, whereas the ciliate heme $c_1$ groups are slightly closer. Furthermore, the distance from Q$_o$ sites (dark gold) to heme $b_L$ is comparable to bovine CIII$_2$ (17).

**(F)** The CIII$_2$ wedged ubiquinone (UQ, transparent gold density) in COB (green) is almost rotationally symmetric, showing planar head group density equally close to the heme $b_L$ molecules (transparent blue). **(G)** Cryo-EM density carved around the UQCRFS1 proteins in CIII$_2$. Head domains show inherent flexibility. **(H)** Location of distal and proximal UQCRFS1 subunits (yellow ribbon) with respect to CI quinone tunnel (black box). The purple box highlights region for focused 3D classification. **(I)** 3D classes I-VIII. UQCRFS1 and UQCRTT3 were fitted to each map and distance measured from Fe$_2$S$_2$ cluster to heme $c_1$. Only classes V and VI display markedly different distances. **(J)** Comparison of B-state class V (left) and C-state class VI (right). B-state UQCRFS1 superposes well with B-state UQCRFS1 from chicken CIII$_2$ (purple, PDB 3BCC). C-state UQCRFS1 superposes almost perfectly with C-state UQCRFS1 from chicken CIII$_2$ (purple, PDB 1BCC). Measured distances are from UQCRFS1 Fe$_2$S$_2$ to heme $c_1$ and $b_L$.

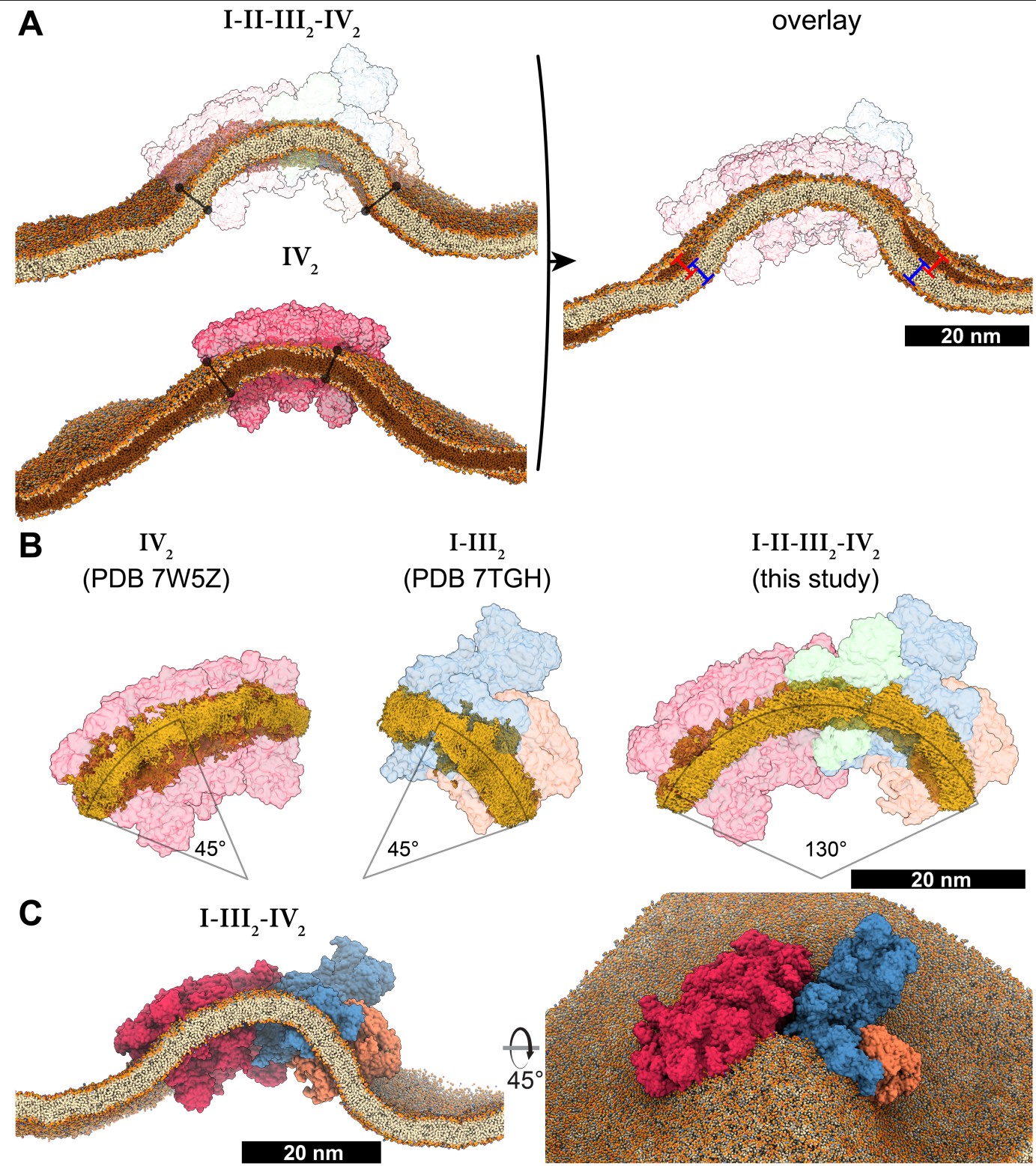

**A**

I-II-III$_2$-IV$_2$

overlay

IV$_2$

20 nm

**B**

IV$_2$
(PDB 7W5Z)

I-III$_2$
(PDB 7TGH)

I-II-III$_2$-IV$_2$
(this study)

45°

45°

130°

20 nm

**C**

I-III$_2$-IV$_2$

20 nm

45°

**Extended Data Fig. 10 | Membrane curvature induction by the supercomplex and subcomplexes. (A)** Left, selected frames of molecular dynamics simulations of the I-II-III$_2$-IV$_2$ supercomplex and CIV$_2$, embedded in a lipid bilayer. Contact sites of protein complexes with sliced plane are shown by black markers. Right, overlay and zoom in of the frames of the I-II-III$_2$-IV$_2$ supercomplex and CIV$_2$. The larger size of the supercomplex and its architecture bends the membrane (blue marker) more than CIV$_2$ alone (red marker). **(B)** Detergent/ saponin belts extracted from the cryo-EM structures of CIV$_2$ (PDB 7W5Z), CI-CIII$_2$ (PDB 7TGH) and of the I-II-III$_2$-IV$_2$ supercomplex. Lengths of curvature arcs when viewed along the membrane tube are indicated. **(C)** Selected frame of the molecular dynamics simulation of the I-III$_2$-IV$_2$ supercomplex (missing CII), displaying a similar curvature to the full supercomplex.

# Reporting Summary

## Statistics

For all statistical analyses, confirm that the following items are present in the figure legend, table legend, main text, or Methods section.

| n/a | Confirmed | |
|---|---|---|
| ☐ | ☒ | The exact sample size (*n*) for each experimental group/condition, given as a discrete number and unit of measurement |
| ☐ | ☒ | A statement on whether measurements were taken from distinct samples or whether the same sample was measured repeatedly |
| ☒ | ☐ | The statistical test(s) used AND whether they are one- or two-sided<br>*Only common tests should be described solely by name; describe more complex techniques in the Methods section.* |
| ☒ | ☐ | A description of all covariates tested |
| ☒ | ☐ | A description of any assumptions or corrections, such as tests of normality and adjustment for multiple comparisons |
| ☒ | ☐ | A full description of the statistical parameters including central tendency (e.g. means) or other basic estimates (e.g. regression coefficient) AND variation (e.g. standard deviation) or associated estimates of uncertainty (e.g. confidence intervals) |
| ☒ | ☐ | For null hypothesis testing, the test statistic (e.g. *F*, *t*, *r*) with confidence intervals, effect sizes, degrees of freedom and *P* value noted<br>*Give P values as exact values whenever suitable.* |
| ☒ | ☐ | For Bayesian analysis, information on the choice of priors and Markov chain Monte Carlo settings |
| ☒ | ☐ | For hierarchical and complex designs, identification of the appropriate level for tests and full reporting of outcomes |
| ☒ | ☐ | Estimates of effect sizes (e.g. Cohen's *d*, Pearson's *r*), indicating how they were calculated |

*Our web collection on statistics for biologists contains articles on many of the points above.*

## Software and code

Policy information about availability of computer code

Data collection
: The datasets were collected EPU 1.9 software on FEI Titan Krios (FEI/Thermofischer) transmission electron microscope operated at 300 keV with a slit width of 20 eV on a GIF quantum energy filter (Gatan). A K2 Summit detector (Gatan) was used at a pixel size of 0.83 Å (magnification of 165,000x) with an exposure rate of 4.26 electrons/pixel/second fractionated over 20 frames. A defocus range of 0.6 to 2.6 μm was used. Tomography data were collected with SerialEM 4.0

Data analysis
: Movie frames were aligned and averaged by global and local motion corrections by the program RELION-3.1. Contrast transfer function (CTF) parameters were estimated by CTFFIND4. Particles were picked and initially 2D classified by RELION 3.1. 2D classification and 3D heterogeneous refinement steps were performed in cryoSPARC v.2.0. The models were manually built with Coot 0.95 and stereochemical refinement was performed using phenix.real_space_refine in the PHENIX 1.19 suite. The final model was validated using MolProbity 4.2. Simulation data analysis is done with GROMACS tools and Visual Molecular Dynamics (VMD 1.9.3 - 1.9.5), Martini3, martinize2 (version 2.6). Tilt series was performed in IMOD 4.11. Subtomogram averaging was performed in PEET 1.9.0.

For manuscripts utilizing custom algorithms or software that are central to the research but not yet described in published literature, software must be made available to editors and reviewers. We strongly encourage code deposition in a community repository (e.g. GitHub). See the Nature Portfolio guidelines for submitting code & software for further information.

## Data

Policy information about availability of data

All manuscripts must include a data availability statement. This statement should provide the following information, where applicable:

- Accession codes, unique identifiers, or web links for publicly available datasets
- A description of any restrictions on data availability
- For clinical datasets or third party data, please ensure that the statement adheres to our policy

> The atomic coordinates were deposited in the RCSB Protein Data Bank (PDB) under accession numbers 8BQS (supercomplex), 8B6F (CI), 8B6G (CII), 8B6H (CIV), 8B6J (CIII). The cryo-EM maps have been deposited in the Electron Microscopy Data Bank (EMDB) under the respective accession numbers EMD-16184, EMD-15865, EMD-15866, EMD-15867, EMD-15868. Subtomogram average have been deposited under EMD-15900.
> The atomic coordinates that were used in this study: 1NTZ [https://www.rcsb.org/structure/1NTZ] (cytochrome bc1), 5IY5 [https://www.rcsb.org/structure/5IY5] (cytochrome c), 5J4Z [https://www.rcsb.org/structure/5J4Z] (ovine supercomplexes)
> Full versions of all gels are provided in the source file. An Excel file has been added. All the data will be publicly available.

## Human research participants

Policy information about studies involving human research participants and Sex and Gender in Research.

| | |
|---|---|
| Reporting on sex and gender | N/A |
| Population characteristics | N/A |
| Recruitment | N/A |
| Ethics oversight | N/A |

Note that full information on the approval of the study protocol must also be provided in the manuscript.

# Field-specific reporting

Please select the one below that is the best fit for your research. If you are not sure, read the appropriate sections before making your selection.

☒ Life sciences  ☐ Behavioural & social sciences  ☐ Ecological, evolutionary & environmental sciences

For a reference copy of the document with all sections, see nature.com/documents/nr-reporting-summary-flat.pdf

# Life sciences study design

All studies must disclose on these points even when the disclosure is negative.

| | |
|---|---|
| Sample size | A total of 26,063 movies were recorded and analyzed. No statistical analyses has been performed. The number of cryo-EM particles in the single dataset collected was the number of particles available. No predetermined sample size was used for other experiments. |
| Data exclusions | For cryo-EM structure determination, particles that were not respiratory supercomplex were discarded by classification, since they cannot contribute to reconstruction. |
| Replication | Cryo-EM structures were successfully obtained from three preliminary datasets. In MD simulations, at least three simulation replicas were performed. Overall, consistent results were obtained from all the different simulation replicas. |
| Randomization | Cryo-EM map resolution estimates by Fourier Shell Correclation were performed using half-maps from random half-sets. N/A to MD simulations. |
| Blinding | MD simulations did not include blinding. N/A to cryo-EM study; raw micrographs or particle images are not categorical data. Particles are randomly assigned into half-sets for image processing; hence no blinding is applicable. |

# Reporting for specific materials, systems and methods

We require information from authors about some types of materials, experimental systems and methods used in many studies. Here, indicate whether each material, system or method listed is relevant to your study. If you are not sure if a list item applies to your research, read the appropriate section before selecting a response.

## Materials & experimental systems

| n/a | Involved in the study |
|-----|----------------------|
| ☒ ☐ | Antibodies |
| ☒ ☐ | Eukaryotic cell lines |
| ☒ ☐ | Palaeontology and archaeology |
| ☒ ☐ | Animals and other organisms |
| ☒ ☐ | Clinical data |
| ☒ ☐ | Dual use research of concern |

## Methods

| n/a | Involved in the study |
|-----|----------------------|
| ☒ ☐ | ChIP-seq |
| ☒ ☐ | Flow cytometry |
| ☒ ☐ | MRI-based neuroimaging |

