## [Peer Review File · Nature]

Editorial Note: the final version of this paper was passed by the referees

Manuscript Title: Structural basis of mitochondrial membrane bending by I-II-III2-IV2 supercomplex

Reviewer Comments & Author Rebuttals

Reviewer Reports on the Initial Version:

Referees' comments:

Referee #1 (Remarks to the Author):

In this manuscript, Mühleip et al. determined the structure of the 5.8 MDa mitochondria respiratory supercomplex from the ciliate *Tetrahymena thermophila* by cryo electron tomography. Advancing on a recent structure of the *Tetrahymena* respiratory supercomplex reported in *Science* (Zhou et al., 2022), this work includes Complex II in the structure. This is the first respiratory supercomplex structure containing Complex II, a significant advance of the work. Additionally, this structure reveals that unlike in any other structure of a respiratory supercomplex reported to date, the mitochondrial inner membrane curves to accommodate the supercomplex. While ATP synthase dimerization is known to stabilize membrane curvature within cristae, this is the first such report of such curvature induction by Complex I-IV supercomplexes, which were previously thought to exclusively reside on the flat part of cristae. Supporting the validity of their high resolution structure, the authors perform cryo-tomography of isolated mitochondria, which revealed cristae with a diameter of 40nm and ATP synthase dimers in helical rows as well as respiratory supercomplexes in a curved membrane confirmation. The authors finally perform molecular dynamics simulations which are additionally supportive of the induced membrane curvature. Together, these data likely explain the tubular cristae architecture observed in *Tetrahymena*.

The structure is quite remarkable and the inclusion of Complex II as well as the finding of membrane curvature induction by Complexes I-IV are major advances. I only have one comment/suggestion. The authors make a conclusion at the end of the abstract that their findings “explain how the architecture of the native supercomplex reflects the functional specialization of bioenergetics by shaping the membrane.” However, this claim is not directly tested by the authors. Is it possible based on their structure to predict a mutation that may disrupt the curvature induction of the supercomplex and test how this affects cristae shape, respiratory growth, and the bioenergetics of *Tetrahymena*?

Referee #2 (Remarks to the Author):

This manuscript reports the cryo-EM SPA structure of a mitochondrial respiratory chain supercomplex from the ciliate *Tetrahymena*. The structure explains some of the major differences to mammalian supercomplexes and reveals new peculiarities. In situ existence of the supercomplex is confirmed by cryo-ET subtomogram averaging (STA) of isolated mitochondrial membranes. The

manuscript's major findings are 1) the supercomplex is formed by all four complexes I – IV. Previously, complex II has not been seen to participate in supercomplexes. 2) The supercomplex has a curved architecture which implies it is embedded into a curved membrane, in line with previous observations that *Tetrahymena* cristae are tubular. 3) The supercomplex is held together in part by subunits encoded by 'split genes' which at their termini are extended in comparison to their mammalian counterpart. These enlarging sequences are the basis of interactions within the supercomplex.

Without doubt this is a very spectacular structure of a gigantic complex, representing the state-of-the-art of structural biology. The quality of the cryo-EM and cryo-ET data appears to be excellent, and the fact that they succeeded to purify this complex to a homogeneity sufficient for SPA proves the authors' excellence. So overall, I recommend publication of this paper, if the authors can address the following concerns.

Major:

1) Recently, a structure of CI + CIII and a structure of CIV from the same organism have been reported (Zhou et al Science 2022), which the authors cite. Zhou et al show that even just CIV alone, as well as the complex of CI+CIII, have a curved architecture that matches and thus possibly induces cristae tubulation. Do the authors see any indication that presence of CII or the assembly into the full complex makes any difference regarding curvature as compared to just CI + CIII or CIV alone? What more do we learn from Muhleip et al regarding membrane curvature induction than what has been shown by Zhou et al? Are there any new insights from the Muhleip et al structure why a high membrane curvature would be beneficial for the electron transfer chain, and the function of the supercomplex? (I agree that a high membrane curvature increases the surface area etc, as said in lines 208-210, but this was clear before – this new structure doesn't contribute to this conclusion). Especially given the title of the manuscript, it seems important to address these questions, or tone down the claim of novelty regarding membrane curvature.

2) Related to the above, lines 197-198: The shape of the supercomplex could also accommodate the supercomplex in high-curvature IMM areas, rather than inducing curvature. It is not straight-forward to distinguish curvature induction from localisation to high curvature.

3) The assignment of the many different co-factors, for example shown in Fig 2F, 3D, E and in Ext Data Fig 8A (heme-c) is astonishing; how is it possible to unambiguously identify all these small molecules at approx. 3 Angstrom resolution? Regarding Ubiquinones, I'm not sure I find assignment of the densities shown in Fig 2F, Ext Data Fig 2 fully convincing. Regarding the hemes in Fig 3D and E: are there (convincingly assignable) densities?

4) I am by no means an expert on evolutionary aspects of ciliates vs. humans, so I wonder how do the authors assume the order of evolutionary events regarding gene fragmentation, expansion and gaining capacity for interactions? (eg. Lines 86, 101, 103) The idea is surely very intriguing, but I am concerned whether this is not a too strong claim unless supported by phylogenetic comparisons.

Minor:

5) Lines 50-51: If I understood Muhleip et al 2016 correctly, which is cited here, the authors previously thought that the tubular cristae can be explained by the ATP synthases assembly alone. Why do they now say it cannot? This also relates to major point 2.

6) Lines 130 and 132: Extended Data Fig 8E is referenced but such a panel doesn't exist.

7) The figures could benefit from some simplification, eg. Fig 2B-G. I appreciate that it is difficult to depict a structure of 130 proteins, but if labels that are not referred to in the text could be removed from the main figures, that would help making the scenes clearer. I have also trouble seeing what Fig 2G is meant to convey, this could also benefit from an improved presentation.

Referee #3 (Remarks to the Author):

The authors present the architecture of a respiratory electron transport chain supercomplex from the ciliate, *Tetrahymena*. Their sample and standard-setting structural analysis revealed the organization of the entire protein complement of the supercomplex, a record-setting number of lipids, including the COQ isoprenoids that are essential for electron transport. Their study covers each of the "subcomplexes" that come together with a unique stoichiometry in this organism by comparison with more familiar animal and fungal architectures. As a piece of structural biology, this work is nothing short of stupendous. The authors assigned a terrific number of proteins, lipids, and small molecules in their density, and each part has been analyzed at the highest level.

Minor comments:

The significance of membrane bending by the ciliate ETC offers an explanation for the cylindrical curvature of the ciliate cristae, we agree. But we did not fully understand the role of membrane curvature in establishing or maintaining proton gradients (author's suggestion). Please consider expanding this argument if space allows.

Referee #4 (Remarks to the Author):

Mühleip et al present a quite interesting cryo-electron microscopy and cryo-tomography structures of the *Tetrahymena thermophila* mitochondria I-II-III2-IV2 supercomplex. Their detailed analysis allows them to reach three major conclusions among a number of very interesting observations:

1. That the supercomplex play a critical role in determining the membrane curvature of the tubular cristae.
2. That CII can participate in the supercomplexes

3. Shows the supercomplex in situ

The manuscript is very sound, well written and provide novel insights on the role of the mitochondrial electron transport chain structural organization. This reviewer has enjoyed reading it and learning from their discoveries. I would like to congratulate the authors for this beautiful piece of work

I however, have few recommendations:

1. I was surprised that no mention to any of the multiple and fundamental original papers from Herman Schagger is made. Even in the methodology for BNGE manuscript 33 is quoted when it represents just a description from the original methodology developed by Herman Schagger. My strong recommendation is that the authors may recognize the seminal contribution from Herman Schagger. By the same token, the authors should remember that the description, analysis, and investigation of the respiratory supercomplexes and its role extended well beyond the cryo-electron microscopy analysis papers.
2. No comment is made on the potential co-existence of free complexes and supercomplexes in *Tetrahymena thermophila* mitochondria. Does your cryo-tomography analysis allowed to postulate the presence of smaller supercomplexes or even free complexes in situ? A comment on that issue may be interesting.
3. In lines 129 to 132 there is a mention to Extended Data Fig. 8E. However, there is no panel 8E in ED Fig 8. If I understood well the authors likely refers to the dashed window coming from panel A. Is this correct?

Author Rebuttals to Initial Comments:

We thank the Reviewers for their kind comments and for taking the time to provide constructive suggestions on how to improve the study, its readability and presentation. We addressed all the requests and followed the valuable suggestions. Particularly, in the revised version, we present new MD simulations that clarify the role of subcomplexes and the supercomplex in the membrane shaping, expand on the discussion to strengthen the argument of membrane bending, clarify figures and tone down some of the previous claims, as requested.

Below is the point-by-point response.

Referee #1 (Remarks to the Author):

In this manuscript, Mühleip et al. determined the structure of the 5.8 MDa mitochondria respiratory supercomplex from the ciliate *Tetrahymena thermophila* by cryo electron tomography. Advancing on a recent structure of the *Tetrahymena* respiratory supercomplex reported in *Science* (Zhou et al., 2022), this work includes Complex II in the structure.

This is the first respiratory supercomplex structure containing Complex II, a significant advance of the work. Additionally, this structure reveals that unlike in any other structure of a respiratory supercomplex reported to date, the mitochondrial inner membrane curves to accommodate the supercomplex. While ATP synthase dimerization is known to stabilize membrane curvature within cristae, this is the first such report of such curvature induction by Complex I-IV supercomplexes, which were previously thought to exclusively reside on the flat part of cristae. Supporting the validity of their high resolution structure, the authors perform cryo-tomography of isolated mitochondria, which revealed cristae with a diameter of 40nm and ATP synthase dimers in helical rows as well as respiratory supercomplexes in a curved membrane confirmation. The authors finally perform molecular dynamics simulations which are additionally supportive of the induced membrane curvature. Together, these data likely explain the tubular cristae architecture observed in *Tetrahymena*.

The structure is quite remarkable and the inclusion of Complex II as well as the finding of membrane curvature induction by Complexes I-IV are major advances. I only have one comment/suggestion. The authors make a conclusion at the end of the abstract that their findings “explain how the architecture of the native supercomplex reflects the functional specialization of bioenergetics by shaping the membrane.” However, this claim is not directly tested by the authors. Is it possible based on their structure to predict a mutation that may disrupt the curvature induction of the supercomplex and test how this affects cristae shape, respiratory growth, and the bioenergetics of *Tetrahymena*?

We removed this potentially misleading claim from the abstract and replaced it by a summarising description of the structural work – “Our findings highlight how the evolution of protein subunits of respiratory complexes has led to the I-II-III₂-IV₂ supercomplex that contributes to the shaping of the bioenergetic membrane, thereby enabling functional specialization.”

In addition, on lines 106-107, where the CI-CIV₂ interface is discussed, we added a clarification “... thus a single mutation is unlikely to disrupt the curvature induction.” that is because the interface involves 25 subunits.

Finally, we added to the study MD simulations that clarify the role of subcomplexes and the supercomplex in the membrane shaping as further described in the reply to Referee #2 below.

Referee #2 (Remarks to the Author):

This manuscript reports the cryo-EM SPA structure of a mitochondrial respiratory chain supercomplex from the ciliate *Tetrahymena*. The structure explains some of the major differences to mammalian supercomplexes and reveals new peculiarities. In situ existence of the supercomplex is confirmed by cryo-ET subtomogram averaging (STA) of isolated mitochondrial membranes.

The manuscript's major findings are 1) the supercomplex is formed by all four complexes I – IV. Previously, complex II has not been seen to participate in supercomplexes. 2) The supercomplex has a curved architecture which implies it is embedded into a curved membrane, in line with previous observations that *Tetrahymena* cristae are tubular. 3) The supercomplex is held together in part by subunits encoded by 'split genes' which at their termini are extended in comparison to their mammalian counterpart. These enlarging sequences are the basis of interactions within the supercomplex.

Without doubt this is a very spectacular structure of a gigantic complex, representing the state-of-the-art of structural biology. The quality of the cryo-EM and cryo-ET data appears to be excellent, and the fact that they succeeded to purify this complex to a homogeneity sufficient for SPA proves the authors' excellence. So overall, I recommend publication of this paper, if the authors can address the following concerns.

Major:

1) Recently, a structure of CI + CIII and a structure of CIV from the same organism have been reported (Zhou et al Science 2022), which the authors cite. Zhou et al show that even just CIV alone, as well as the complex of CI+CIII, have a curved architecture that matches and thus possibly induces cristae tubulation. Do the authors see any indication that presence of CII or the assembly into the full complex makes any difference regarding curvature as compared to just CI + CIII or CIV alone? What more do we learn from Muhleip et al regarding membrane curvature induction than what has been shown by Zhou et al? Are there any new insights from the Muhleip et al structure why a high membrane curvature would be beneficial for the electron transfer chain, and the function of the supercomplex? (I agree that a high membrane curvature increases the surface area etc, as said in lines 208-210, but this was clear before – this new structure doesn't contribute to this conclusion).

Especially

given the title of the manuscript, it seems important to address these questions, or tone down the claim of novelty regarding membrane curvature.

To address this comment, we performed additional analyses and toned down the claim. The new data is described on lines 210-221. Particularly, to test whether CIV₂ alone induces similar membrane curvature, we performed MD simulations, and the data are shown in SI Fig. 7A-B. The results of the simulations reveal that although in the vicinity of subcomplexes, a similar membrane behaviour is observed as for the I-II-III₂-IV₂ supercomplex, distal to protein surface, the membrane starts to regain a different architecture. This suggests that subcomplexes can induce membrane bending to some extent, but the complete tubular-like architecture requires the full I-II-III₂-IV₂ supercomplex.

To further compare the point of the membrane curvature with Zhou et al, we extracted the detergent/amphipol belt from the cryo-EM structure of CIV₂ (PDB 7W5Z) and CI-CIII₂ (PDB 7TGH) and illustrated with that of the I-II-III₂-IV₂ supercomplex. The data are shown in SI Fig. 7C-E. The lengths of curvature arcs when viewed along the membrane tube differ considerably. The information has been added to the text on lines 215-217.

Finally, we performed MD simulations of I-III₂-IV₂ supercomplex (lacking CII), and the results in SI Fig. 7C suggest that the membrane may still wrap around in a similar fashion as in the full

supercomplex. Thus, CII does not directly contribute to the curved architecture, but likely stabilises the supercomplex, and we now added this information on lines 217-221. We also clarified in Fig. 2F that CI, CII and CIV are connected via the membrane and luminal regions. The related claims have been toned down, and we removed from the abstract the sentence "... explain how the architecture of the native supercomplex reflects the functional specialization of bioenergetics by shaping the membrane."

It is our humble opinion that the work presented here, from single particle to tomography and validation with MD simulations, goes well beyond what has previously been suggested from the separate structures of dissociation products of the full supercomplex.

2) Related to the above, lines 197-198: The shape of the supercomplex could also accommodate the supercomplex in high-curvature IMM areas, rather than inducing curvature. It is not straight-forward to distinguish curvature induction from localisation to high curvature.

To support our conclusions that protein architecture plays an important role in initiating and maintaining membrane curvature, we performed additional MD simulations of pure lipid bilayer consisting of the same type and concentration of lipids as in the MD setup for the supercomplex. The coarse-grained molecular dynamics simulation of pure lipid bilayer is presented in SI Video 2. The membrane bilayer in the absence of protein remains flatter, suggesting that the assembly of a rigid protein architecture, such as the supercomplex can induce membrane bending.

Similarly, mutational analyses of mitochondrial ATP synthase in yeast have shown that the curved dimeric membrane region actively generates highly curved membrane regions, which are absent in monomeric mutant strains (refs 33, 34), suggesting a causative relationship, and not a passive diffusion into pre-existing curvature regions. We added this information on lines 232-233.

3) The assignment of the many different co-factors, for example shown in Fig 2F, 3D, E and in Ext Data Fig 8A (heme-c) is astonishing; how is it possible to unambiguously identify all these small molecules at approx. 3 Angstrom resolution? Regarding Ubiquinones, I'm not sure I find assignment of the densities shown in Fig 2F, Ext Data Fig 2 fully convincing. Regarding the hemes in Fig 3D and E: are there (convincingly assignable) densities?

*Regarding the density assigned to ubiquinone in CII (previously Fig. 2F), it is found in the Q_P site that is universally conserved, including coordinating residues, thus to clarify we added "conserved" on line 130. In addition, we prepared SI Fig 4 to illustrate conservation of this quinone binding site between the *T. thermophila* CII and the porcine homolog including the interacting tryptophan residue of the binding pocket.*

Regarding the hemes in CIII (Fig. 3D, E), they are also conserved, and this information has now been added on lines 159-161. In addition, we added three panels in SI Figure 5D, showing density map around conserved hemes b_L , b_H and c_1 as they are found in the CIII₂.

The heme-C group in complex-II (Ext Data Fig 8A) agrees well with the flat, square cryo-EM density. Furthermore, the distinct heme coordination environment of the surrounding protein structure is in strong agreement with this assignment, including two axial histidines (H260, H317), that coordinate the iron metal ion and a cystein (C321) showing continuous density indicative of a covalent bond. Thus, both cryo-EM map and protein structure support ligand assignment. In addition, the visible redox spectra recorded on the purified supercomplex clearly reveal the presence of at least one B- or C-type heme group in addition to the canonical cytochrome b_H , b_L and c_1 , supporting our assignment.

Finally, the assignments of ubiquinones in CIV₂ (Ext Data Fig 2D-F) are made based on almost identical shapes of headgroup densities that match the protruding methoxy groups in each of the two CIV monomers, and the tails display the typical protrusions of the isoprenoid groups. The assignment is further supported by a ubiquinone assignment in the density of the T. thermophila ATP synthase, ref 12.

4) I am by no means an expert on evolutionary aspects of ciliates vs. humans, so I wonder how do the authors assume the order of evolutionary events regarding gene fragmentation, expansion and gaining capacity for interactions? (eg. Lines 86, 101, 103) The idea is surely very intriguing, but I am concerned whether this is not a too strong claim unless supported by phylogenetic comparisons.

We toned down the claim by deleting “subsequently” on line 83 and changing “evolutionary” to “putative” on line 103. The mechanistic aspect described here is unlikely to be picked up from a sequence analysis or phylogenetic comparisons, due to highly deviated sequences of extensions. This also explains why it wasn’t reported prior to the current study, and thus further highlights the value of the structural analysis.

Minor:

5) Lines 50-51: If I understood Muhleip et al 2016 correctly, which is cited here, the authors previously thought that the tubular cristae can be explained by the ATP synthases assembly alone. Why do they now say it cannot? This also relates to major point 2.

*Rephrased to “...was previously explained by the helical row assembly of ATP synthase.”
Moreover, we now more clearly distinguish the roles of the ATP synthase and supercomplex in lines 222-226.*

6) Lines 130 and 132: Extended Data Fig 8E is referenced but such a panel doesn’t exist.

Fixed the typo to “8A”.

7) The figures could benefit from some simplification, eg. Fig 2B-G. I appreciate that it is difficult to depict a structure of 130 proteins, but if labels that are not referred to in the text could be removed from the main figures, that would help making the scenes clearer. I have also trouble seeing what Fig 2G is meant to convey, this could also benefit from an improved presentation.

We have tried to improve the presentation of Fig 2. Specifically, we removed panel 2F, increased the size of panels 2B, 2C, 2D, remade panel 2G, so that it clearly shows the three complexes connected in the lumen. The new view shows CI, CII, CIV₂ in flat lighting and features the triangle cleft with the interconnecting subunits.

Referee #3 (Remarks to the Author):

The authors present the architecture of a respiratory electron transport chain supercomplex from the ciliate, Tetrahymena. Their sample and standard-setting structural analysis revealed the organization of the entire protein complement of the supercomplex, a record-setting number of lipids, including the COQ isoprenoids that are essential for electron transport. Their study covers each of the "subcomplexes" that come together with a unique stoichiometry in this organism by comparison with more familiar animal and fungal architectures. As a piece of structural biology, this work is nothing

short of stupendous. The authors assigned a terrific number of proteins, lipids, and small molecules in their density, and each part has been analyzed at the highest level.

Minor comments:

The significance of membrane bending by the ciliate ETC offers an explanation for the cylindrical curvature of the ciliate cristae, we agree. But we did not fully understand the role of membrane curvature in establishing or maintaining proton gradients (author's suggestion). Please consider expanding this argument if space allows.

We thank the Referee for giving us the opportunity to clarify the argument, which is now done on lines 222-235. We observe for the first time that in addition to the ATP synthase long dimer rows setting the membrane outer perimeter, the presence of a second membrane-shaping complex restricts the tube diameter to 40 nm, and together they generate a relatively small luminal volume. The reduced luminal volume will also locally increase the proton motive force that drives ATP synthesis by the ATP synthase following a correlation between the compartment size, electrochemical gradient and membrane potential that has been previously suggested. and references have been added to the text.

Referee #4 (Remarks to the Author):

Mühleip et al present a quite interesting cryo-electron microscopy and cryo-tomography structures of the *Tetrahymena thermophila* mitochondria I-II-III2-IV2 supercomplex. Their detailed analysis allows them to reach three major conclusions among a number of very interesting observations:

1. That the supercomplex play a critical role in determining the membrane curvature of the tubular cristae.
2. That CII can participate in the supercomplexes
3. Shows the supercomplex in situ

The manuscript is very sound, well written and provide novel insights on the role of the mitochondrial electron transport chain structural organization. This reviewer has enjoyed reading it and learning from their discoveries. I would like to congratulate the authors for this beautiful piece of work I however, have few recommendations:

1. I was surprised that no mention to any of the multiple and fundamental original papers from Herman Schägger is made. Even in the methodology for BNGE manuscript 33 is quoted when it represents just a description from the original methodology developed by Herman Schägger. My strong recommendation is that the authors may recognize the seminal contribution from Herman Schägger. By the same token, the authors should remember that the description, analysis, and investigation of the respiratory supercomplexes and its role extended well beyond the cryo-electron microscopy analysis papers.

We thank the Referee for raising this important point. We now cite the seminal work by Hermann Schägger, the discovery of mitochondrial respiratory supercomplexes using a newly developed native electrophoresis method, as the first reference on lines 41-42, thereby crediting the biochemical work that preceded structure determination of these complexes.

2. No comment is made on the potential co-existence of free complexes and supercomplexes in *Tetrahymena thermophila* mitochondria. Does your cryo-tomography analysis allowed to postulate the presence of smaller supercomplexes or even free complexes in situ? A comment on that issue may be interesting.

The cryo-tomography analysis was based on the presence of the matrix arm of CI, and thus it might provide a somewhat limited insight about a possibility of presence of smaller subcomplexes, however it does suggest that the I-II-III₂-IV₂ supercomplex is the most abundant form, and we added the comment on lines 197-199.

3. In lines 129 to 132 there is a mention to Extended Data Fig. 8E. However, there is no panel 8E in ED Fig 8. If I understood well the authors likely refers to the dashed window coming from panel A. Is this correct?

Thank you. Changed to Extended Data Fig. 8A.

Reviewer Reports on the First Revision:

Referees' comments:

Referee #1 (Remarks to the Author):

The authors have fully addressed my concerns and I recommend the manuscript for publication.

Referee #2 (Remarks to the Author):

The revised manuscript is much improved and the authors have addressed previous concerns mostly satisfactory.

There are still some reservations regarding the interpretation of membrane curvature induction by the supercomplex. The new MD data are state-of-the-art and provide additional evidence of membrane curvature potentially being induced by the supercomplex structure. More precisely, what these new results show is that the supercomplex has the capability to deform a bilayer. Whether the supercomplex is the major or sole driver of membrane curvature in *Tetrahymena* cristae is not answered. In fact, it is very difficult if not impossible to discern for a “real” cellular membrane which component is the major driver of curvature, as it is likely the combination of different aspects including the ensemble of membrane proteins, protein local concentration, lipid composition and mechanical forces that will determine membrane shape. The authors are thus advised to be more careful with their phrasing, especially in the title of the paper and in the abstract (particularly in the second sentence). It is appropriate to say that the supercomplex might be the driver or stabiliser of the tubular cristae curvature (or, as the author write in the abstract, that the supercomplex actively contributes to the membrane curvature induction) but not that the authors show or demonstrate that membrane curvature is provided by the supercomplex. This paper is nevertheless truly spectacular!

Further, minor points:

- Cryo-ET data collection and processing is not referred to in the reporting summary. This should be added.
- The authors should consider depositing their subtomogram averaging (STA) map in the EMDB as well, in addition to the SPA structures.
- The resolution estimate (FSC) of the STA map should be included, e.g in the supplementary file.
- Concerning the MD simulations, the author state that “during production runs, the (...) harmonic constraints on the backbone beads were applied”. Is this correct? If so, I would change “applied” to “maintained” and the authors should justify this decision (is the complex unstable without constraints? Or simply to focus on the lipid rearrangement while keeping the protein fixed?). Otherwise, if the constraints were “removed” during production, the author should discuss how they dealt with the secondary structure of the protein complex (did they use an elastic network?).

Referee #3 (Remarks to the Author):

In this manuscript, Mühleip et al. determined the structure of the 5.8 MDa mitochondria respiratory supercomplex from the ciliate *Tetrahymena thermophila* by cryo-electron microscopy and tomography. This is the first respiratory supercomplex structure containing Complex II and reveals that unlike in any other structure of a respiratory supercomplex reported to date, the mitochondrial inner membrane curves to accommodate the shape of the supercomplex. Supporting the validity of their high-resolution single-particle structure, the authors also performed cryo-tomography of isolated mitochondria. This analysis revealed cristae with a diameter of 40nm and ATP synthase dimers in helical rows, as well as respiratory supercomplexes in highly-curved membrane tubules. Finally, the authors performed molecular dynamics simulations that support the claim that the shape of the supercomplex influences membrane curvature. Together, these data likely explain the tubular cristae architecture observed in *Tetrahymena* and yield the most comprehensive structural understanding of the many proteins, lipids, and quinone molecules underlying energy transduction in this ciliate.

Accept!

Referee #4 (Remarks to the Author):

The authors have satisfactorily answer the concerns raised in my first review. I have no further comments.

Author Rebuttals to First Revision:

Referee #2:

The revised manuscript is much improved and the authors have addressed previous concerns mostly satisfactory.

There are still some reservations regarding the interpretation of membrane curvature induction by the supercomplex. The new MD data are state-of-the-art and provide additional evidence of membrane curvature potentially being induced by the supercomplex structure. More precisely, what these new results show is that the supercomplex has the capability to deform a bilayer. Whether the supercomplex is the major or sole driver of membrane curvature in *Tetrahymena* cristae is not answered. In fact, it is very difficult if not impossible to discern for a "real" cellular membrane which component is the major driver of curvature, as it is likely the combination of different aspects including the ensemble of membrane proteins, protein local concentration, lipid composition and mechanical forces that will determine membrane shape. The authors are thus advised to be more careful with their phrasing, especially in the title of the paper and in the abstract (particularly in the second sentence). It is appropriate to say that the supercomplex might be the driver or stabiliser of the tubular cristae curvature (or, as the author write in the abstract, that the supercomplex actively contributes to the membrane curvature induction) but not that the authors show or demonstrate that membrane curvature is provided by the supercomplex. This paper is nevertheless truly spectacular!

We made the following changes:

- The second sentence of the Abstract changed to: "Here we show that in ciliates, a supercomplex containing all four respiratory chain components, contributes to membrane curvature induction."

- The last sentence of the manuscript changed to: "Thus, our findings show how respiratory supercomplexes together with other factors can organise the architecture of the bioenergetic membrane, providing a mechanism for enabling its functional specialization."

Further, minor points:

- Cryo-ET data collection and processing is not referred to in the reporting summary. This should be added.

- Added.

- The authors should consider depositing their subtomogram averaging (STA) map in the EMDB as well, in addition to the SPA structures.

We deposited the data and added: "Subtomogram average have been deposited under EMD-15900."

- The resolution estimate (FSC) of the STA map should be included, e.g in the supplementary file.

Included in Extended Data Fig. 1E "Fourier Shell correlation of the subtomogram average indicating a resolution of 28 Å".

- Concerning the MD simulations, the author state that "during production runs, the (...) harmonic constraints on the backbone beads were applied". Is this correct? If so, I would change "applied" to "maintained" and the authors should justify this decision (is the complex unstable without constraints? Or simply to focus on the lipid rearrangement while keeping the protein fixed?). Otherwise, if the constraints were "removed" during production, the author should discuss how they dealt with the secondary structure of the protein complex (did they use an elastic network?).

Correct, and we have changed "applied" to "maintained" in the text. Our decision to keep constraints during production runs was in part guided by the aim to observe how lipid bilayer assembles around a protein preserved in structural conformation. Second, our preliminary tests to simulate the large supercomplex without any constraints showed that the structure of the protein was less stable, caused by the local unfolding and loosened binding of individual subunits. This is in part related to the widely acknowledged and difficult problem of optimization of protein-protein interactions in coarse-grained force fields like Martini. On the other hand, the usage of elastic network and other similar approaches (that are used to maintain optimal protein-protein interactions) was found to be technically challenging because of the large size of the supercomplex and presence of large number of protein subunits. Therefore, the usage of harmonic restraints in production runs is justified as our focus was to observe the behavior of large membrane patch around a protein that is maintained in a structurally and biologically relevant conformation.